# A genome and gene catalog of the aquatic microbiomes of the Tibetan Plateau

Mingyue Cheng [1,6], Shuai Luo[2,3,6], Peng Zhang [2,4,6], Guangzhou Xiong [1], Kai Chen [2], Chuanqi Jiang[2], Fangdian Yang[2,4], Hanhui Huang[1], Pengshuo Yang[1], Guanxi Liu[1], Yuhao Zhang[1], Sang Ba[4], Ping Yin [3], Jie Xiong [2,5] ✉, Wei Miao [2,3,4,5] ✉ & Kang Ning [1] ✉

The Tibetan Plateau supplies water to nearly 2 billion people in Asia, but climate change poses threats to its aquatic microbial resources. Here, we construct the Tibetan Plateau Microbial Catalog by sequencing 498 metagenomes from six water ecosystems (saline lakes, freshwater lakes, rivers, hot springs, wetlands and glaciers). Our catalog expands knowledge of regional genomic diversity by presenting 32,355 metagenome-assembled genomes that de-replicated into 10,723 representative genome-based species, of which 88% were unannotated. The catalog contains nearly 300 million non-redundant gene clusters, of which 15% novel, and 73,864 biosynthetic gene clusters, of which 50% novel, thus expanding known functional diversity. Using these data, we investigate the Tibetan Plateau aquatic microbiome's biogeography along a distance of 2,500 km and >5 km in altitude. Microbial compositional similarity and the shared gene count with the Tibetan Plateau microbiome decline along with distance and altitude difference, suggesting a dispersal pattern. The Tibetan Plateau Microbial Catalog stands as a substantial repository for high-altitude aquatic microbiome resources, providing potential for discovering novel lineages and functions, and bridging knowledge gaps in microbiome biogeography.

The Tibetan Plateau, spanning over 2.5 million square kilometers and averaging an altitude of over 4500 m, is the highest and largest plateau globally, with its ice fields containing the largest reserve of freshwater outside the polar regions, thus known as the "third pole[1–3]". Its unique topographical features render the Tibetan Plateau a primary source of numerous streams and rivers, functioning as a "water tower" that sustains surrounding regions and provides a reliable water supply to almost 2 billion people[4,5].

The diverse water ecosystems within the Tibetan Plateau, encompassing glaciers, wetlands, hot springs, lakes, and rivers, harbor a rich diversity of aquatic microorganisms[6–11], including those from uncultivated new lineages[6], and exhibit substantial biosynthetic potential[6]. The Tibetan Plateau microbiome may demonstrate distinct patterns in adaptation to the extreme environmental conditions of the Tibetan Plateau, such as drastic temperature fluctuations, low oxygen concentrations, low pressure, and intense ultraviolet radiation[6,12].

[1]Key Laboratory of Molecular Biophysics of the Ministry of Education, Hubei Key Laboratory of Bioinformatics and Molecular Imaging, Center of Artificial Intelligence Biology, Department of Bioinformatics and Systems Biology, College of Life Science and Technology, Huazhong University of Science and Technology, Wuhan, China. [2]Institute of Hydrobiology, Chinese Academy of Sciences, Wuhan, China. [3]National Key Laboratory of Crop Genetic Improvement, Hubei Hongshan Laboratory, Huazhong Agricultural University, Wuhan, China. [4]Laboratory of Tibetan Plateau Wetland and Watershed Ecosystem, College of Science, Tibet University, Lhasa, China. [5]Key Laboratory of Breeding Biotechnology and Sustainable Aquaculture, Chinese Academy of Sciences, Wuhan, China. [6]These authors contributed equally: Mingyue Cheng, Shuai Luo, Peng Zhang. ✉e-mail: xiongjie@ihb.ac.cn; miaowei@ihb.ac.cn; ningkang@hust.edu.cn

However, the Tibetan Plateau is particularly vulnerable to the effects of climate change, including global warming[6,13,14], which is expected to cause substantial water loss in the region over this century (>10 billion metric tons per year lost since 2002[14]). Consequently, there is an urgent need for the cataloging and protection of Tibetan Plateau's aquatic microbial resources to ensure their preservation for future generations.

In recent years, advancements in sequencing technologies and computational methods within metagenomics have facilitated the reconstruction of genomes from previously uncultivated microorganisms. This has led to the discovery of functional potential within a diverse range of microbial resources on a large scale[6,15–17]. For metagenome-assembled genomes (MAGs) in Tibetan Plateau metagenomes, Liu et al. successfully recovered 2358 MAGs from 85 snow, ice, and cryoconite metagenomes of glaciers[6]. Wei et al. also contributed to this field by recovering 75 MAGs from two soil and water metagenomes of a saline lake[18], while Yun et al. obtained 200 MAGs from two soil samples of wetlands[19]. Additionally, Hu et al. recovered 278 MAGs from 69 water and sediment metagenomes of wetlands and rivers[20]. Notably, among these studies, only the Tibetan Plateau glacier catalog has been systematically established by Liu et al.[6]; the other studies did not have the specific goal of establishing a resource. It is crucial to emphasize the imperative need to catalog microbial resources in various ecosystems such as rivers, due to their essential role in the hydrological cycle and their connections to human societies. Wetlands are also of paramount importance, given that approximately 80% of global wetland resources are degrading or disappearing[21].

Here we explored the structure and the assemblage of the aquatic microbial communities on the Tibetan Plateau and established a comprehensive genome and gene catalog known as the Tibetan Plateau Microbial Catalog (TPMC), by investigating 498 metagenomic samples (Supplementary Fig. 1). The TPMC comprises 32,355 MAGs that de-replicated into 10,723 representative genome-based species, along with an extensive repository of nearly 300 million non-redundant gene clusters, 20% of which were novel, and 73,864 biosynthetic gene clusters, with 50% being novel. Furthermore, our research extended to the biogeography of the Tibetan Plateau microbiome across a 2500-km transect, revealing a discernible microbial dispersal pattern. The TPMC is, to our knowledge, the largest and most comprehensive repository of aquatic microbial resources on the Tibetan Plateau. This catalog sheds light on the taxonomic and functional adaptations of aquatic microbial communities, and also holds the potential to facilitate the preservation of the global microbiome reservoir.

## Results

### Microbial community composition, assemblage, and networks exhibited unique patterns across diverse water ecosystems

To examine the microbial community patterns across water ecosystems, we investigated 498 metagenomic samples from both the central plateau of the Tibetan Plateau (abbreviated as Tibet, $n = 356$) and the northeastern border of the Tibetan Plateau (Qilian Mountains-Qinghai Lake, abbreviated as Qilian, $n = 142$). These samples encompassed diverse water ecosystems, including saline lakes ($n = 104$), freshwater lakes ($n = 72$), rivers ($n = 108$), hot springs ($n = 76$), wetlands ($n = 132$), and glaciers ($n = 6$) (Fig. 1a). The analysis revealed that the microbial community composition was dominated by the phyla Pseudomonadota, Bacteroidota, and Actinomycetota (Supplementary Fig. 2a). Notably, *Cyanobium usitatum* was among the most abundant species in saline lakes, freshwater lakes, rivers in Tibet, and saline lakes in Qilian (Supplementary Fig. 2b). *Acinetobacter johnsonii* featured prominently in rivers in Qilian, while *Polynucleobacter duraquae* dominated wetlands in both Tibet and Qilian. Additionally, *Tepidimonas fonticaldi* was the most abundant species in hot springs in Tibet.

Utilizing principal coordinates analysis (PCoA) on Bray–Curtis distances of genera profiles (Fig. 1b), we observed distinct microbial community patterns. Notably, microbial communities in freshwater lakes and saline lakes were markedly different from those in the other four ecosystems ($P < 0.001$). Furthermore, samples from freshwater lakes exhibited greater dispersion within the same region compared to samples from the other ecosystems, indicating a high degree of geographical heterogeneity. Additionally, our analysis revealed that microbial communities in saline lakes exhibited more pronounced distinctions between Tibet and Qilian compared to wetlands and rivers. This observation suggests that the saline lake microbial communities were more sensitive to the regional effects.

To decipher the underlying factors contributing to these microbial differences, we conducted an in-depth examination of microbial assemblages using the neutral community model (NCM). This model, grounded in neutral theory, assists in deducing community assembly processes by evaluating the presence of stochasticity through the diversity pattern[22]. As shown in Fig. 1c and Supplementary Fig. 2c, most microbial communities displayed a limited fit to the NCM model ($R^2 < 0.3$), except for Qilian saline lakes ($R^2 = 0.565$), with river microbial communities exhibiting the lowest fit ($R^2 = 0.021$). This suggested a minimal contribution of stochastic processes in community assembly, highlighting the prominent role of environmental selective pressures in shaping microbial community diversity. Specifically, factors such as dissolved inorganic phosphorus and $NO_2^{2-}$ content in rivers, total nitrogen and $NO_3^{3-}$ content, optical dissolved oxygen in freshwater, and $SO_4^{2-}$ and $Br^-$ content, temperature, pH, and salt concentration in saline lakes exhibited significant associations with variability in microbial community composition ($P < 0.05$ and $FDR < 0.1$, Supplementary Data 2.1). Microbial ecology networks (MENs) further revealed distinct stability patterns among these microbial communities (Fig. 1d, Supplementary Fig. 2d, and Supplementary Data 2.2). Tibet rivers and freshwater lakes presented the highest vulnerability, with $Vul = 0.12$ and $Vul = 0.11$, respectively. Notably, Qilian rivers had the lowest vulnerability ($Vul = 0.04$), but a significant increase was observed after 50% of the nodes were removed from the network, surpassing the vulnerability levels of other ecosystems in Qilian. This suggested that MENs of Qilian rivers might be more sensitive to abrupt ecological variation. Furthermore, Tibet saline lakes and Tibet hot springs exhibited the highest relative modularity in MENs, with $RM = 2.06$ and $RM = 2.51$, respectively, indicating stronger biointeractions within the microbial module. In contrast, Tibet wetlands showed a negative RM of −0.33, indicating the microbes within a module were less interacted and had more interactions with microbes from other modules. These findings underscored that different Tibetan Plateau water ecosystems on the Tibetan Plateau have developed unique microbial community patterns in response to extreme environmental conditions.

### TPMC provides a vast genomic resource including over 33,000 metagenome-assembled genomes from diverse water ecosystems

We next set out to systematically catalog the microbial genomes within these unique microbiome resources across diverse water ecosystems in Tibetan Plateau. We performed metagenomic assembly and binning on the 498 metagenomes and recovered a total of 32,355 MAGs (Supplementary Data 1). These MAGs met or exceeded the medium-quality criteria outlined in the Minimum Information about a Metagenome-Assembled Genome (MIMAG) standard[23], with mean completeness of 78.1% and mean contamination of 2.6%. Of these MAGs, 25,017 exhibited a quality score above 50 (defined as completeness − (5 × contamination)), and 2024 were assigned as high quality, featuring the presence of the 23 S, 16 S, and 5 S rRNA genes and at least 18 tRNAs (Fig. 2a, b and Supplementary Data 1). The assembly sizes of these MAGs showed no significant difference ($P = 0.3$, Supplementary Data 2.3) between Tibet (average: 2.5 Mb,

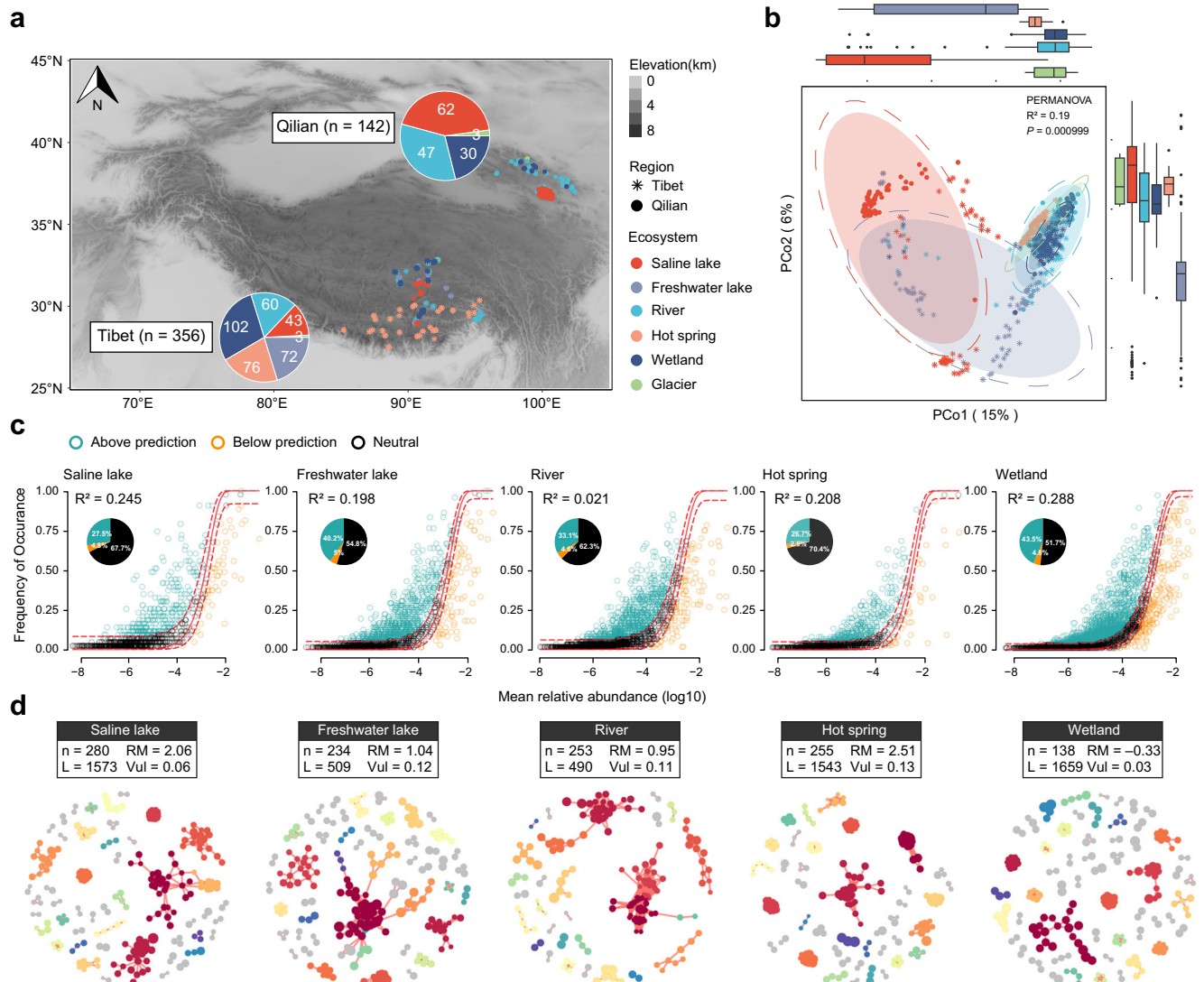

**Fig. 1 | Composition, assemblage, and networks of the Tibetan Plateau aquatic microbial communities. a** Geographic distribution of metagenomic samples from six water ecosystems (Saline lakes, Freshwater lakes, Rivers, Hot springs, Wetlands, and Glaciers) in two large-scale regions (Tibet and Qilian). **b** Microbial community compositions (*n* = 498 independent samples) plotted on a PCoA plot based on Bray–Curtis distance of the genera profiles produced by the MetaPhlAn (version 4.0.1)[37]. The statistical significance of the dispersions between groups is calculated by PERMANOVA (R² and *P*). The shaded ellipses represent the 85% confidence interval, and the dotted ellipse borders represent the 90% confidence interval. Boxplots show the sample distributions against the PCo 1 and PCo 2 axes. Boxes represent the interquartile range between the first and third quartiles and the line inside represents the median. Whiskers denote the lowest and highest values within the 1.5×interquartile range from the first and third quartiles, respectively. **c** Fit of

the NCMs of microbial community assemblage across five water ecosystems in Tibet. The solid red lines indicate the best fit to the NCM, and the dashed red lines represent 95% confidence intervals around the model prediction. Genera that occur more or less frequently than those predicted by the NCM are shown in different colors. R² indicates the fit of the model. **d** The MENs of five water ecosystems in Tibet were constructed based on Spearman correlations of genera relative abundances produced by MetaPhlAn (version 4.0.1)[37]. Modules with ≥2 nodes are shown in different colors, and smaller modules are shown in gray. Details of network topological attributes are listed in Supplementary Data 2.2. The NCMs and MENs in Qilian are shown in Supplementary Fig. 2c and d, respectively. PCOA, principal coordinates analysis; PERMANOVA, Permutational multivariate analysis of variance; NCM neutral community model, MEN molecular ecological network.

interquartile range (IQR) = 1.6–3.1 Mb) and Qilian (2.5 Mb, 1.6–3.1 Mb). However, the GC content was slightly higher (*P* = 1.2 × 10⁻¹³) in Tibet (52.4%, 42.8–62.1%) than in Qilian (51.5%, 41.8–60.4%). Glacier (2.77 Mb, 2.1–3.4 Mb) and wetland microbiomes (2.6 Mb, 1.7–3.2 Mb) showed the highest assembly sizes, while wetland microbiomes showed the lowest GC content (51.1%, 41.4–61.1%). Furthermore, we clustered 32,355 MAGs into 10,723 representative genome-based species, using a whole-genome average nucleotide identity (ANI) threshold of 95% and an aligned fraction (AF) threshold of 30% (Fig. 2c and Supplementary Data 1). Delineation of these species was genome-based and the term "species" will be used for conciseness.

We sought to assess the novelty of these representative TPMC species through a comparative analysis with representative genomes from the Genome Taxonomy Database (GTDB release R214)[24], recognized as the most comprehensive reference genome database for Bateria and Archaea. Additionally, we compared these MAGs with representative genomes from Earth's Microbiomes catalog (GEM)[16] that shared many types of water ecosystems but differed in geography positions with TPMC. Moreover, we conducted a comparison with representative genomes from the Tibetan Glacier Genome catalog (TG2G)[6], which currently represents another genomic catalog of the Tibetan Plateau microbiome. Furthermore, representative genomes of the TARA ocean[25] were also recruited for

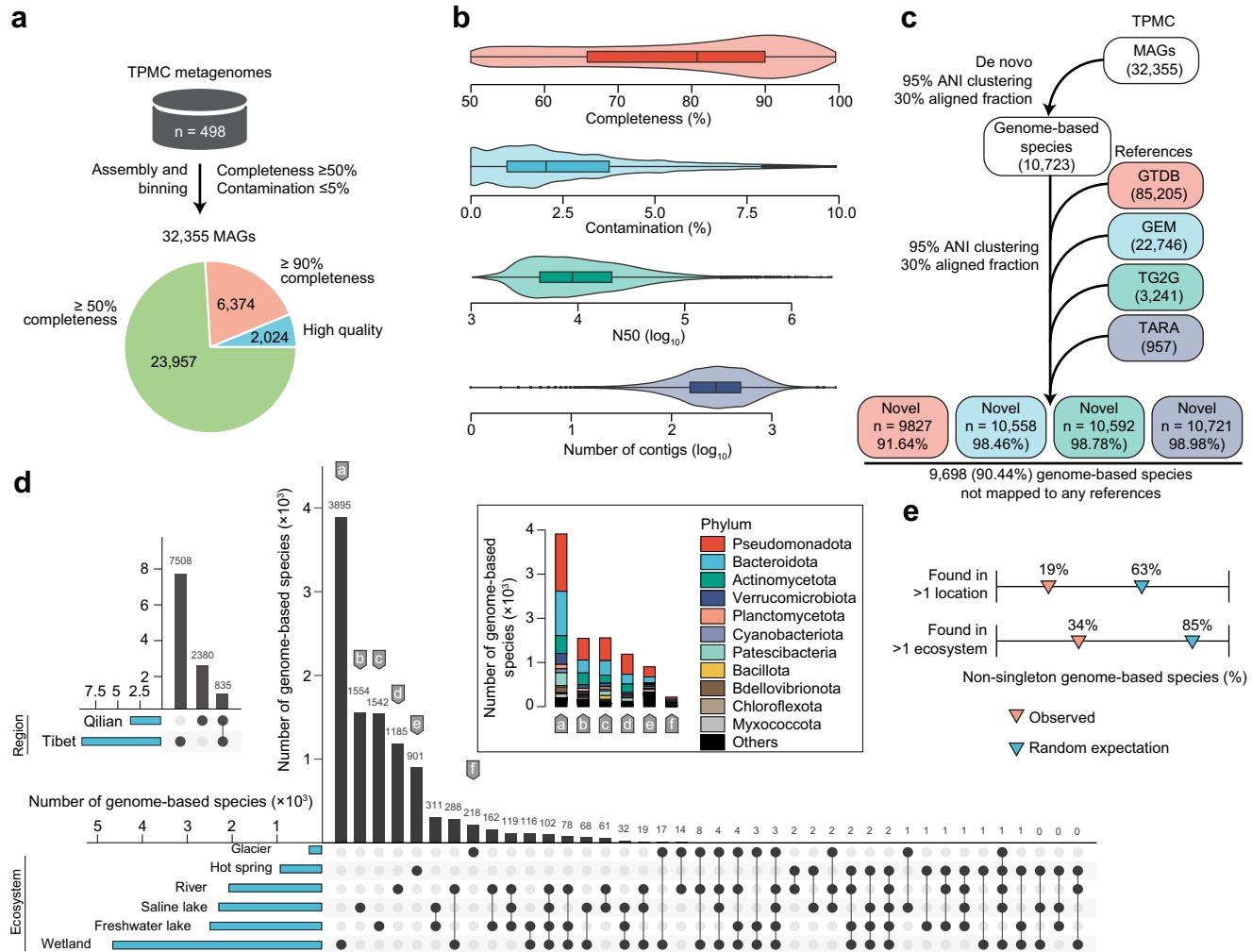

**Fig. 2 | Environmental and geographical distribution of metagenome-assembled genomes and their species-level clustering. a** A total of 32,355 MAGs recovered from 498 geographically and environmentally diverse microbial community samples in the TPMC. All MAGs have a completeness of at least 50% and a contamination level of less than 5%. **b** Distribution of quality metrics for the MAGs ($n = 32,355$). Boxes represent the interquartile range between the first and third quartiles and the line inside represents the median. Whiskers denote the lowest and highest values within the 1.5× interquartile range from the first and third quartiles, respectively. **c** TPMC MAGs are clustered into 10,723 genome-based species, based on 95% ANI and 30% AF. Representative genomes from GTDB R214 ($n = 85,205$),

GEM ($n = 22,746$), TG2G ($n = 3241$), and TARA ($n = 957$) were then included in the clustering to identify the novelty of the TPMC species. **d** Geographical and environmental distribution of TPMC species. Region or water ecosystem labels are assigned to species that are clustered by MAGs from diverse regions or water ecosystems. The species that contained MAGs recovered solely from a specific ecosystem were annotated by letters. **e** The majority of TPMC species with >1 MAG are restricted to individual water ecosystems and regions (Tibet or Qilian). MAG metagenome-assembled genome, ANI average nucleotide identity, AF alignment fraction.

comparison. The 10,723 TPMC species demonstrated substantial novelty, with 91.64%, 98.46%, 98.78%, and 99.98% exhibiting low sequence identity to 85,205 GTDB genomes, 22,746 GEM genomes, 3241 TG2G genomes, and 957 TARA genomes, respectively (Supplementary Data 2.4). A total of 9698 species (90.44%) did not map to any of these catalog genomes, of which 6335 passed a quality score >50, and 465 met high-quality criteria. In particular, 219 out of 245 species from the TPMC glaciers did not map to the TG2G genomes. Given that many types of Tibetan Plateau water ecosystems such as saline lakes, freshwater lakes, hot springs, and wetlands were also covered by GEM, the high level of novelty observed suggested that the unique geographic and environmental conditions of the Tibetan Plateau may endow the TPMC with unique microbes and metabolic potential. TPMC stands as a comprehensive and novel resource of the Tibetan Plateau microbiome across diverse water ecosystems. It expands the previous GTDB database by 11.5% (9827 novel genomes compared to 85,205 references) and the GEM database by 46.4% (10,558 novel genomes compared to 22,746

references), thus largely contributing to the completion of the global microbiome reservoir.

Next, we investigated the distributions of the species across Tibetan Plateau water ecosystems. The vast majority of species contained MAGs exclusively recovered in a single region ($n = 9888$, 92.2%) or water ecosystem ($n = 9295$, 86.7%) (Fig. 2d). Notably, 36.3% ($n = 3895$) of species were exclusively recovered in the wetland, a significantly higher proportion than those exclusively recovered in other water ecosystems (218 (2.0%)–1554 (14.5%)). We further examined non-singleton species ($n = 4188$) that contained more than one MAG. Most of these were still region-specific ($n = 3353$, 81%) or water ecosystem-specific ($n = 2760$, 66%) (Fig. 2e). These results implied the region/biome-dependent discovery potential of microbial species in TPMC resources, especially in the Tibet wetland.

Utilizing taxonomic annotations from the Genome Taxonomy Database (GTDB release R214)[24], we identified a wide spectrum of taxonomic diversity within the TPMC, covering 83 known phyla, 186 known classes, 470 known orders, 952 known families, 1835 known

genera, and 993 known species (Supplementary Data 1). Subsequently, we constructed a phylogeny of the 10,723 TPMC species utilizing 120 concatenated bacterial marker genes, and highlighted the top 11 phyla containing more than 1% of the total MAGs (Fig. 3a and Supplementary Fig. 4a). Among these phyla, the MAG catalog was predominantly composed of Pseudomonadota ($n = 10,183$, 31.5%), Bacteroidota ($n = 7551$, 23.3%), Actinomycetota ($n = 6163$, 19.0%), and Verrucomicrobiota ($n = 1993$, 6.2%). The dominance of Pseudomonadota (synonym Proteobacteria), Bacteroidota, and Actinomycetota was consistent with previous studies in Tibetan Plateau saline lakes and rivers[18,20]. However, we recovered more MAGs from Verrucomicrobiota, and it took the place of Bacillota (synonym Firmicutes), which dominated previous studies[18,20]. Conversely, Bdellovibrionota ($n = 568$, 1.8%), Chloroflexota ($n = 430$, 1.3%), and Myxococcota ($n = 329$, 1.0%) had the lowest count of MAGs. In addition, the hot spring manifested a paucity of lineages from the dominated phyla, while the bulk of the lineages belonging to the Chloroflexota could be recovered in the hot spring (Fig. 3a). Moreover, besides bacterial genomes ($n = 32,224$, 99.6%), we recovered 131 (0.4%) archaeal genomes, dominated by phyla Thermoproteota ($n = 65$), Halobacteriota ($n = 24$), and Nanoarchaeota ($n = 16$), and genus *Methanothrix* ($n = 11$) (Supplementary Data 1).

Furthermore, our analysis revealed that a considerable number of TPMC species ($n = 9384$, 87.5%) remained unclassified by GTDB (Supplementary Data 1). Among these unclassified species, the majority belonged to the phyla Pseudomonadota ($n = 3003$, 32.0%), Bacteroidota ($n = 2098$, 22.4%), and Actinomycetota ($n = 1224$, 13.0%), as well as the genera *Flavobacterium* ($n = 287$, 3.1%), *Planktophila* ($n = 167$, 1.8%), and *Rhodoluna* ($n = 149$, 1.6%). Notably, the genera *Rhodoluna* and *Planktophila* had only been annotated with three and six species in the current database[26], respectively, while our catalog has expanded such novel microbial lineages.

## TPMC possessed substantial functional potential with nearly 300 million non-redundant genes

Besides cataloging genomic resources, we extensively explored and cataloged the functional potential within the TPMC. We predicted and clustered 522,671,245 open reading frames (ORFs) into 296,289,678 non-redundant gene clusters, referred to as unigenes. The TPMC showcased vast and largely unexplored functional potential, as evidenced by the increasing number of unigenes with sampling depth (Supplementary Fig. 5a). Notably, among all the ecosystems, the Tibet wetland was the most functionally promising, contributing the greatest number of unigenes compared to other ecosystems at the same

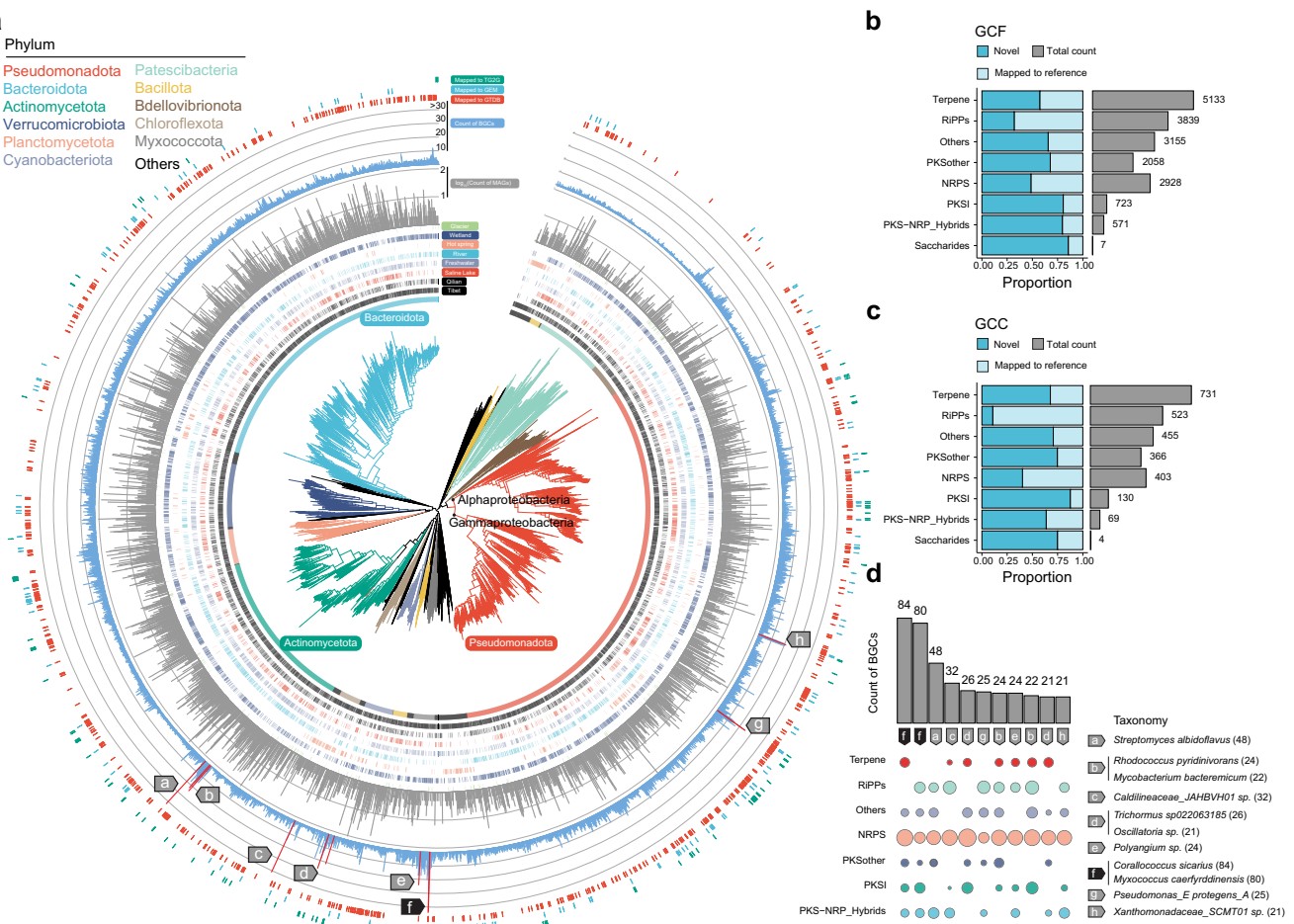

**Fig. 3 | Novelty and phylogenomic distribution of the metagenome-assembled genomes and their biosynthetic potential. a** A phylogenetic tree built for all the TPMC genome-based species ($n = 10,723$) based on a concatenated alignment of 120 universally distributed bacterial single-copy genes and placement of each MAG in the GTDB-Tk reference tree. Two classes of the phylum Pseudomonadota are marked in the tree. The color of the branches indicates the phylum of the species. The outer layers indicate, for each species, the region and water ecosystem labels based on its MAGs, the count of MAGs, the largest count of BGCs among its MAGs, and the novelty (compared to the representative genomes of GTDB, GEM, and TG2G catalogs, respectively). **b, c** BGCs ($n = 73,864$) are clustered into GCFs ($n = 18,414$) and GCCs ($n = 2681$), with their novelty and the total number across different types presented. **d** MAGs with more than 20 BGCs are labeled in **a**, and the count of BGCs across different types of these MAGs is presented. MAG metagenome-assembled genome, BGC biosynthetic gene cluster, GCF gene cluster family, GCC gene cluster clan.

sampling depth or sequencing depth (Supplementary Fig. 3 and Supplementary Fig. 5a). Moreover, only a limited number of unigenes were found to be shared among regions ($n = 15,496,510$, 5.2%) or water ecosystems ($n = 28,093$, 0.01%) (Supplementary Fig. 3). This observation indicated that distinct gene sets were present in various regions and water ecosystems, suggesting an adaptive response to the diverse and challenging environmental conditions.

The non-redundant gene catalog underwent taxonomic and functional annotations using the NR, UniRef50, and Swiss-Prot databases, resulting in annotations for 82.4%, 79.8%, and 35.1% of the unigenes, respectively. Functional annotations using COG, KEGG, CAZy, GO, CARD, and VFDB databases[27–30] were also performed, leading to annotations for 66.9%, 46.1%, 1.7%, 17.2%, 0.01%, and 8.9% of the unigenes, respectively (Supplementary Fig. 5b, c and Supplementary Data 2.5). A total of 46,670,736 (15.8%) unigenes remained unannotated by any of the databases, thus designated as novel genes. Across all ecosystems, a predominant proportion of the genes were related to cell wall biosynthesis (e.g., RfaB, WcaG) and inorganic ion transport and metabolism (e.g., CirA, FepA), indicating the enhanced capability of Tibetan Plateau microbes to fortify their cell walls and utilize inorganic ions for structural integrity and resistance to harsh conditions. Moreover, the prevalent presence of the biosynthesis of the secondary metabolites pathway (ko01110) suggested a vast biosynthetic potential within the TPMC, which might serve as a valuable biotechnological resource and provide insights into microbial adaptation to Tibetan Plateau extreme environments.

Particularly, as the water supply to almost 2 billion people[4], the distribution of virulence factors (VFs) in Tibetan Plateau could pose a threat to human health. To evaluate this risk, we identified a total of 26,443,206 potential VFs in the TPMC gene catalog based on VFDB[28]. More than 80% of these factors were categorized as immune modulation (28.2%), nutritional/metabolic factor (21.6%), adherence (10.1%), effector delivery system (9.2%), motility (8.1%), and regulation (7.02%) (Supplementary Fig. 6a and Supplementary Data 2.5). The highest number of VFs were observed in genomes of Pseudomonadota ($n = 1,255,248$), Bacteroidota ($n = 580,452$), and Actinomycetota ($n = 330,708$) (Supplementary Fig. 6b). Additionally, we studied the VF gene count distributions across water ecosystems and regions (Supplementary Fig. 6c, d). Notably, wetland samples exhibited VF0082 (Type IV pili, Adherence) and VF0840 (MymA operon, Immune modulation) as dominant VFs. VF0082 is involved in transcriptional regulation and chemosensory pathways controlling the twitching motility of the pili, while VF0840 modulates small multidrug resistance (SMR) antibiotic efflux pump. In contrast, VF0465 (Capsule, Immune modulation) was exclusively annotated in genes from saline lake samples, responsible for synthesizing a specific monosaccharide that helps protect bacteria from the host's innate immune response. Another VF, VF0091 (Alginate, Biofilm), predominantly annotated in genes from river and wetland samples, facilitates biofilm formation and acts as an adhesin, making it challenging for host phagocytes to ingest and eliminate bacteria. These findings highlight the varied distribution patterns of VFs across diverse water ecosystems in the Tibetan Plateau. This knowledge could serve as a valuable resource for VF monitoring, aiding in the mitigation of potential risks to human health.

## TPMC possessed broad and diverse biosynthetic potential with over 73,000 biosynthetic gene clusters

The TPMC genes showcased a remarkable functional potential in the pathway of secondary metabolites (Supplementary Data 2.5). To assess the richness of such biosynthetic capabilities within the TPMC, we employed antiSMASH[31] on all 32,355 MAGs to predict a total of 73,864 biosynthetic gene clusters (BGCs), which were categorized into eight groups (Supplementary Data 2.6). Terpenes emerged as the most abundant class of BGCs, constituting 43.0% ($n = 31,734$) of the total BGCs. This dominance was consistent across the majority of bacterial

phyla and water ecosystems (Supplementary Fig. 4), aligning with the previously reported prevalence of terpenes in the diversity of Tibet glacier BGCs[6]. Moreover, we identified a total of 11,772 (16.0%) ribosomally synthesized and post-translationally modified peptide (RiPP) clusters from 52 phyla, 7044 (9.6%) non-ribosomal peptide synthetase (NRPS) gene clusters from 31 phyla, 2,043 (2.8%) polyketide synthase (PKS) I clusters from 18 phyla, 1859 (2.5%) PKS–NRPS hybrid clusters from 14 phyla (Supplementary Data 2.7). More than half of the BGCs ($n = 41,657$, 56.4%) were identified in phyla Pseudomonadota ($n = 24,987$, 33.8%) and Bacteroidota ($n = 16,670$, 22.6%). Furthermore, the freshwater lakes in Tibet and the saline lakes in Qilian exhibited the highest biosynthetic potential in the TPMC, with 21,154 (28.6%) and 15,959 (21.6%) BGCs identified, respectively. In contrast, the hot spring and glacier in Tibet presented the lowest biosynthetic potential in the TPMC, with only 2384 (3.2%) and 179 (0.2%) BGCs identified, respectively (Supplementary Data 2.8).

To assess the novelty of the biosynthetic potential within the TPMC, we clustered the 73,864 BGCs into 18,414 gene cluster families (GCFs) and 2681 gene cluster classes (GCCs) (Supplementary Data 2.6). We then compared the GCFs or GCCs with the BiG-SLICE database[32], which contains 1,225,071 BGCs from 209,206 publicly available microbial genomes (Supplementary Data 2.9). A GCF or GCC was considered novel if less than 20% or 40% of its BGCs were mapped to BiG-SLICE (Methods), respectively. In the TPMC, we identified 10,128 (55.0%) novel GCFs and 1,471 (54.9%) novel GCCs (Supplementary Data 2.10 and 2.11). Most BGC types had more than 50% novel GCFs or GCCs, including terpenes, PKS other, PKSI, and PKS-NRP hybrids clusters (Fig. 3b, c). These findings suggested that Tibetan Plateau bacteria might be capable of producing various secondary metabolites with unique structures and functions to cope with harsh environmental conditions. The broad, diverse, and novel biosynthetic potential in the TPMC could greatly expand the current global BGC resources.

We next delved into the metagenomic diversity of the biosynthetic potential across diverse water ecosystems in the Tibetan Plateau. We estimated the metagenomic abundances of BGCs, GCFs, and GCCs by mapping the biosynthetic genes to the metagenomic reads of all samples (Methods). Among all water ecosystems and regions, rivers and wetlands exhibited the highest diversity of BGCs, GCFs, and GCCs, while hot springs from Tibet displayed the lowest diversity ($P < 0.01$, Supplementary Fig. 7). We then clustered the samples based on their GCC profiles and obtained four groups: cluster 1 consisted mainly of the river and wetland samples; cluster 2 mainly consisted of hot spring samples; cluster 3 mainly consisted of freshwater lake samples; and cluster 4 mainy consisted of saline lake samples (Supplementary Fig. 8). The unique BGC compositions observed in Qilian Saline lakes may be linked to their distinct microbial compositions compared to other ecosystems (Fig. 1b). Furthermore, their microbial assemblage demonstrated a better fit to the neutral theory, suggesting a lesser degree of environmental intervention in microbiome assemblage (Supplementary Fig. 2c). Further investigations are warranted to elucidate the mechanisms behind the formation of these unique BGC compositions in Qilian Saline lakes. Notably, Terpenes were the most abundant BGC type in all four clusters, despite their varying proportions across water ecosystems and regions. These results indicated that bacteria from different water ecosystems and regions presented distinct biosynthetic profiles, suggesting their specific ecological adaptations and interactions.

Furthermore, we identified 11 BGC-rich species that contained at least one MAG with >20 BGCs (Fig. 3d and Supplementary Data 2.12) and found their BGCs were enriched in NRPs and RiPPs clusters. Notably, two representative novel MAGs from the phylum Pseudomonadota with the highest number of BGCs, classified as *Myxococcus caerfyrddinensis* (80 BGCs, Genome size: 13 Mb) and *Corallococcus sicarius* (84 BGCs, Genome size: 10 Mb), were phylogenetically close

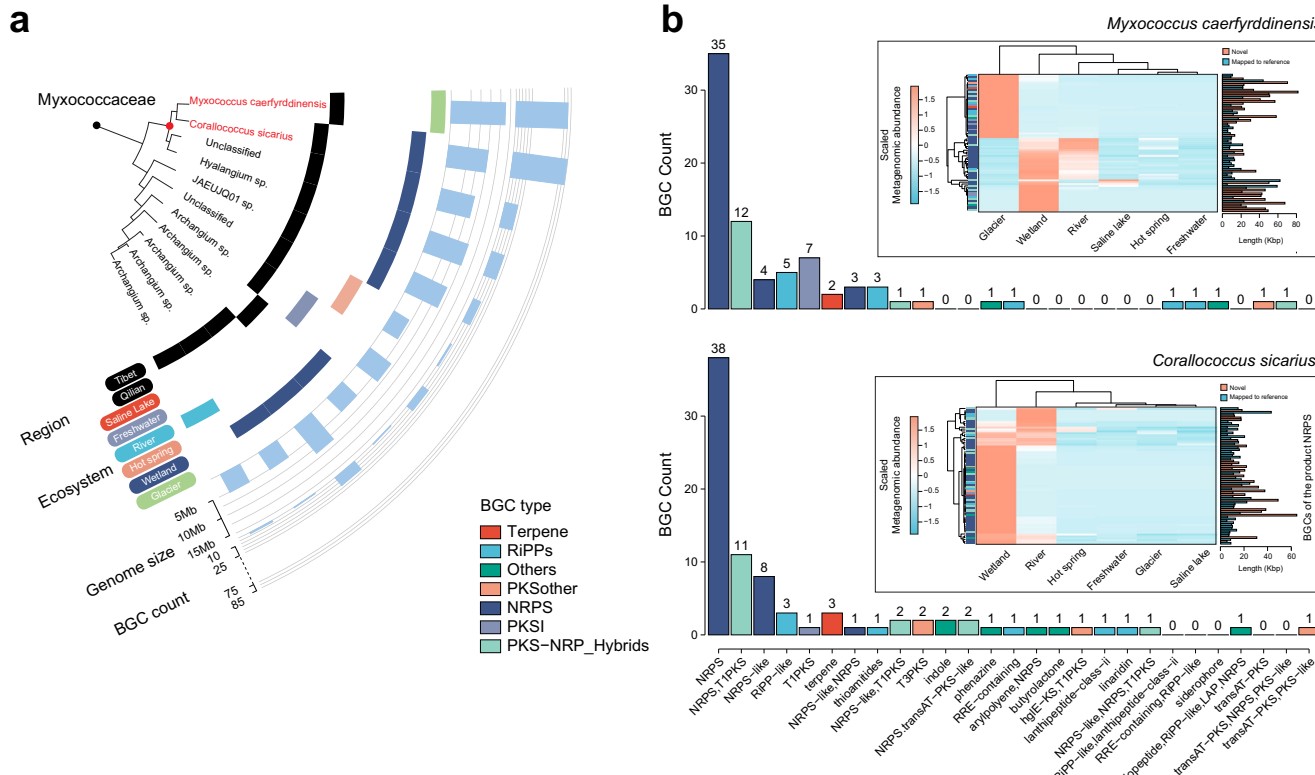

**Fig. 4 | Biosynthetic potential of two BGC-rich species across regions and water ecosystems. a** A phylogenetic sub-tree of family Myxococcaceae is pruned from the tree of the TPMC genome-based species. The outer layers indicate, for each species, the region and water ecosystem labels based on its MAGs, the assembly size of the encompassed MAGs with the largest count of BGCs. The red color highlights the two species of interest: *Myxococcus caerfyrddinensis* and *Corallococcus sicarius*. **b** The BGC count of *Myxococcus caerfyrddinensis* and *Corallococcus sicarius* across different BGC products. Heatmaps show the scaled metagenomic abundance of the BGCs across water ecosystems, with the gene length presented. MAG metagenome-assembled genome, BGC biosynthetic gene cluster, NRPS nonribosomal peptides.

but had distinct geographic distributions. The former was recovered in the Qilian glaciers, while the latter was recovered in Tibet wetlands (Fig. 4a). These two MAGs exhibited a high biosynthetic capability of the product NRPS with at least 35 NRPS-encoding BGCs. Moreover, certain of their BGCs could be mapped to and were even more metagenomically enriched in other water ecosystems (Fig. 4b), such as the wetlands and rivers in Tibet (for *Myxococcus caerfyrddinensis*), and rivers in Qilian (for *Corallococcus sicarius*). Nearly half of the BGCs of the *Myxococcus caerfyrddinensis* genome (*n* = 38, 47.5%) and BGCs of the *Corallococcus sicariu* genome (*n* = 35, 41.7%) are novel. The longest novel BGCs of *Myxococcus caerfyrddinensis* genome synthesized NRPS and T1PKS, with 12 core modules (82,306 bp), and the longest novel BGCs of *Corallococcus sicarius* genome synthesized NRPS with 3 core modules (64,575 bp) (Supplementary Fig. 9). These BGC-rich species could be strong candidates for exploring the underlying mechanisms of microbes in adaptation to the extreme environment of the Tibetan Plateau.

**Microbiome biogeography across a 2500-km transect in China**

The establishment of TPMG has enabled us to explore the microbiome biogeography across a 2500-km transect that spans from the Tibetan Plateau to the east coast of China (Fig. 5a). The transect has a ladder-like topography based on altitude: the Tibetan Plateau itself as the first step of the ladder (ladder 1), characterized by altitudes exceeding 4 km; the second step (ladder 2), ranging between 1 and 2 km in altitude; and the third step (ladder 3), with altitudes below 0.5 km (Fig. 5a). To investigate the microbiome biogeography across the transect, we collected additional freshwater samples from the ladder 2 (*n* = 15) and the ladder 3 (*n* = 19), as well as river samples from the ladder 2 (*n* = 40)

and the ladder 3 (*n* = 24), along with wetland samples from the ladder 2 (*n* = 11). These 109 samples were then combined with 498 samples from the Tibetan Plateau, spanning a geographical distance of over 2500 km, and an altitude difference of over 5 km. Using the Tibetan Plateau samples as reference coordinates, we compared samples within the Tibetan Plateau, as well as between the Tibetan Plateau and the other ladders. Our analysis revealed that the microbial similarity between samples within the same water ecosystem decreased with increasing geographical distance (R = −0.49, *P* < 2.2 × 10⁻¹⁶; Fig. 5b; Supplementary Fig. 10), and with increasing altitude difference (R = −0.47, *P* < 2.2 × 10⁻¹⁶; Fig. 5c; Supplementary Fig. 11). Specifically, for river samples, the most abundant phylum Pseudomonadota in the Tibetan Plateau exhibited a decreasing trend along with the transect, while the other abundant phylum Verrucomicrobia showed an increasing trend (Fig. 5d).

Furthermore, through the clustering of genes from all these 607 samples, we constructed a non-redundant gene catalog comprising a total of 329,568,659 unigenes, representing one of the largest gene catalogs in a single microbiome study[6,16,33,34]. At the functional level, our findings demonstrated that the number of shared genes between samples within the same water ecosystem declined with increasing geographical distance (R = −0.55, *P* < 2.2 × 10⁻¹⁶; Fig. 5e; Supplementary Fig. 10) and increasing altitude difference (R = −0.53, *P* < 2.2 × 10⁻¹⁶; Fig. 5f; Supplementary Fig. 11). Interestingly, these variation trends exhibited a continuous manner in the river ecosystem, where we observed a range up to 1,358,996 shared unigenes spanning 2403 km in distance and 5.4 km in height. In addition to the geographical factors, environmental factors also influenced Tibetan Plateau microbiome biogeography, such as the optical dissolved oxygen,

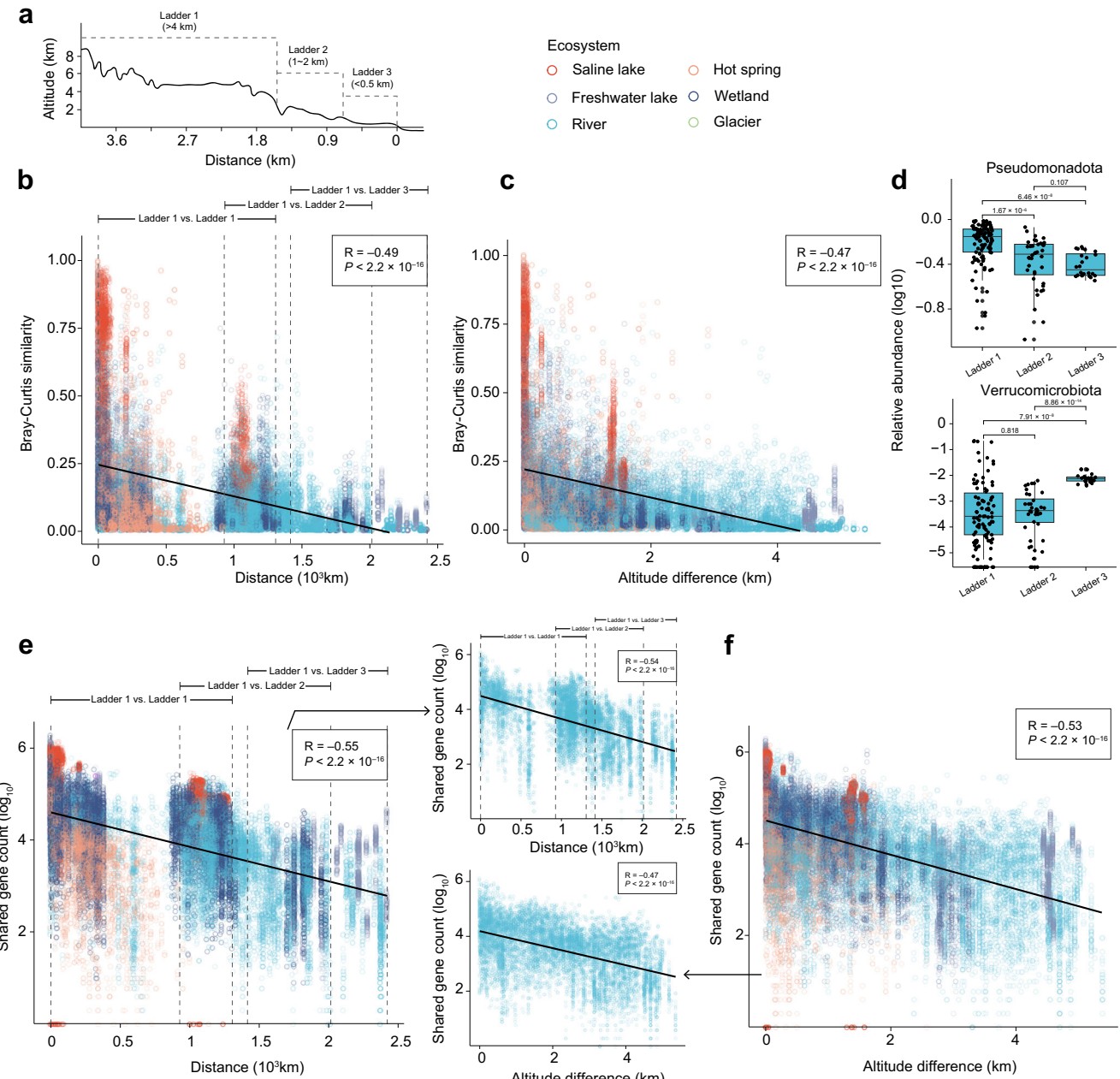

**Fig. 5 | The microbiome biogeography across a 2,500-km transect. a** The terrain profile of China along 36 degrees north latitude to show the ladder-like topography. The microbial species profiles become less similar between samples in the same water ecosystem, as the distance (**b**) and altitude (**c**) difference increase. Spearman's rank correlation coefficient (R) and statistical significance (P) are calculated. **d** The abundance variation of two phyla in the river samples from across a 2500-km transect (Ladder 1, $n = 108$; Ladder 2, $n = 40$; Ladder 3, $n = 24$). Each dot represents the relative abundance ($\log_{10}$) of the phylum of each sample, produced by the MetaPhlAn (version 4.0.1)[37]. The statistical significance of the abundance differences between steps of the ladder is calculated by the two-sided Mann–Whitney–Wilcoxon test. Boxes represent the interquartile range between the first and third quartiles and the line inside represents the median. Whiskers denote the lowest and highest values within the 1.5× interquartile range from the first and third quartiles, respectively. The number of the shared genes decreases between samples in the same water ecosystem, as the distance (**e**) and altitude (**f**) difference increase. Spearman's rank correlation coefficient (R) and statistical significance (P) are calculated. In all scatter plots, each dot represents a pairwise comparison in 1) microbial similarity quantified by the Bray–Curtis distance of species profiles produced by MetaPhlAn (version 4.0.1)[37]; 2) shared gene count; 3) geographical distance; and 4) altitude, which are conducted on samples within the Tibetan Plateau, as well as between the Tibetan Plateau and the others.

temperature, and nitrogen content ($P < 2.2 \times 10^{-16}$; Supplementary Fig. 12). The effects of the geographic and environmental factors were intertwined and not easily disentangled: Geographic variation (latitude, longitude, and altitude) was correlated with taxonomic composition (partial two-sided Mantel test accounting for environmental variation between samples: $R_{\text{Geo|Env}} = 0.15$, $P = 0.001$), which was slightly higher than the correlations of the environmental variation with taxonomic composition ($R_{\text{Env|Geo}} = 0.11$, $P = 0.001$). These findings revealed a potential dispersal pattern in microbiome biogeography. For various water ecosystems situated at greater geographical distances and altitude differences from the Tibetan Plateau, their microbiomes may exhibit fewer compositional and functional similarities with the Tibetan Plateau microbiome, underscoring the potential role of the Tibetan Plateau as a "microbial tower."

## Discussion

The genome and gene catalog of Tibetan Plateau, comprising 32,355 medium- and high-quality MAGs and 296,376,624 non-redundant genes, stands as the largest and most comprehensive resource to date for capturing the genomic and gene diversity across Tibetan Plateau's water ecosystems. The TPMC significantly expands the number of species present in the GTDB by 11.5% and the NCBI non-redundant gene set by approximately fifty million in number. Additionally, alongside the elucidation of microbial communities in terms of structure profiling, network, assembly processes, and interactions with environmental parameters, this catalog provides a large-scale illustration of the enrichment and distribution of aquatic microbial diversity at gene, genome, taxonomic, and functional levels, showcasing their adaptation to the extreme environmental conditions of the Tibetan Plateau. The TPMC catalog also serves as a vast resource of biosynthetic potential, containing a total of 73,864 BGC, of which over 50% are novel, greatly expanding the reservoir of enzymology and natural products for biotechnological application. Furthermore, in the context of Tibetan Plateau suffering substantial water loss due to climate change, TPMC is the first attempt to resolve and catalog the aquatic microbial resources across a range of water ecosystems in Tibetan Plateau, facilitating the preservation of the Tibetan Plateau water resource.

This study for the first time revealed the biogeography of aquatic microbiome across a 2500-km transect in China. Our findings demonstrated that as the distance and altitude difference from the Tibetan Plateau increased, microbial species profiles became less similar to those of the Tibetan Plateau microbiome, and the number of shared functional genes with the Tibetan Plateau microbiome decreased. Moreover, these variations followed a continuous manner in river samples and showed no abrupt curve change. We believe this continuous variation in biogeography might be partly attributed to the dispersal of the microorganisms and their genes that "spread" from the Tibetan Plateau water tower. More samples from East and Southeast Asia could help us to separate this effect from the environmental heterogeneity[35], and to confirm our speculation that the Tibetan Plateau was not only a water tower but also a "microbial tower" serving as a source of microbial diversity and functional potential. Furthermore, for biogeography on a global scale[35,36], our Tibetan Plateau genome catalog exhibited only a small proportion of shared genomes (1.5%) with the GEM genome catalog[16], underscoring Tibetan Plateau's unique microbial resources and their distinct responses to extreme environmental stresses.

In conclusion, the TPMC represents a comprehensive compilation of microbiome resources from diverse water ecosystems in the Tibetan Plateau, which can serve as a valuable reference for comparative studies conducted under extreme altitude and environmental conditions. The catalog enhances our understanding of microbial biogeographic patterns in the region, and it could contribute to biotechnological applications in enzymology, environmental protection, and healthcare.

## Methods

### Sample collection

A total of 498 water samples were collected from multiple sampling stations in the Tibetan Plateau from 2019 to 2021, including the stations in the central plateau of the Tibetan Plateau (abbreviated as Tibet, $n = 356$) and stations in the northeastern border of the Tibetan Plateau, namely the Qilian Mountains-Qinghai Lake (abbreviated as Qilian, $n = 142$). Moreover, to investigate the biogeography of the Tibetan Plateau microbiome, an additional 109 water samples were collected from other sampling stations, including stations of Chi Shui River (26 river samples), Shen Nong Jia (14 river samples, 15 freshwater lake samples, and 11 wetland samples), Bao An Lake (19 freshwater lake samples), and Yangtze River (24 river samples).

For sampling, 5 L of surface water (10–30 cm depth) was collected at each station using a water sampler. The collected water was filtered using a 200 μm filter to remove insoluble impurities, large organisms, aquatic plants, and other debris. The filtered water sample was then vacuum-filtered onto a 0.22 μm PC membrane (Millipore, Billerica, MA, USA). The membrane was immediately placed in aluminum foil, deposited in vials containing Silica gel desiccant, subjected to desiccation, and stored at –80 °C in the laboratory for 3–7 days before subsequent metagenomic sequencing. Environmental parameters of the whole water samples were measured at each station. The longitude, latitude, and altitude were measured using a portable GPS locator (NAVA, F70, China). Various factors, including dissolved oxygen saturation, chlorophyll, depth, fluorescent dissolved organic matter, oxidation-reduction potential, pressure (PSIA), salt concentration, blue-green algae phycocyanin concentration, total amount of dissolved solids, total amount of suspended solids, temperature, and pH were measured using a YSI-EXO2 (YSI, Yellow Springs, OH, USA). Heavy metal concentrations (Ni, Cu, Zn, As, Cd, Pb, Cr, Mn, and Fe) were assessed using a portable heavy metal detector (AVVOR 9000, Canada). Chemical oxygen demand, biological oxygen demand, total phosphorus, total nitrogen, nitrate nitrogen, nitrite nitrogen, dissolved inorganic phosphorus, and ammonia nitrogen of water samples were measured according to the Chinese national standard method (SEPA, 2002a). Total organic carbon was analyzed using an Elementar vario TOC select TOC analyzer (Langenselbold, Germany). Turbidity was measured using a high-precision turbidity analyzer (HI98703, HANNA, Italy). Cation concentrations ($Si^{2+}$, $Na^+$, $K^+$, and $Ca^{2+}$) in water samples were measured using inductively coupled plasma atomic emission spectrometry (ICP-AES) in the laboratory. The concentration anions ($F^-$, $Cl^-$, $Br^-$, and $SO_4^{2-}$) were measured using an ion chromatograph (930 Compact IC Flex, Switzerland).

### DNA Extraction and metagenome sequencing

The cetyltrimethylammonium bromide (CTAB) method was used to extract environmental DNA from the filter membrane. First, the membrane was cut into small pieces using scissors (cut more than eight times). Subsequently, the cut membrane was placed in a 1.5 mL EP tube, and 800 μL of CTAB extraction buffer (2% CTAB, 100 mmol/L Tris-HCl (pH 8.0), 20 mmol/L EDTA, 1.4 mol/L NaCl) was added for mixing. Then, grinding beads were added to it, and after ground using a grinder Tissuelyser-192L (Jingxin, Shanghai, China, 55 Hz, 5 min), it was heated at 70 degrees Celsius for 10 minutes in a water bath to ensure sufficient DNA extraction. Following the extraction of DNA using separation methods, an equal volume of chloroform: isoamylalcohol (24:1) was used to remove proteins. The DNA degradation degree and potential contamination were monitored on 1% agarose gels. The DNA purity was determined using the NanoPhotometer spectrophotometer (IMPLEN, CA, USA), and the DNA concentration was measured using the Qubit dsDNA Assay Kit in Qubit 2.0 Fluorometer (Life Technologies, CA, USA). One microgram of qualified DNA was used to construct the library. For high-quality DNA, the Covaris (Covaris Inc, Woburn, MA, USA) instrument was used to fragment DNA into approximately 350 bp fragments, followed by end repair and addition of "A" tails to the 3' end. The Illumina sequencing adapters were then ligated to both ends of the library DNA using T4 ligase. After library quality control, the libraries were sequenced on the Illumina MGISEQ 2000 Platform (China National GeneBank, Shenzhen, China). Sequencing adapters of the paired-end reads were removed using the SOAPnuke filter (version 2.1.5).

### Microbial community diversity and assembly process

The taxonomic abundance profile of the TPMC microbial community for diversity analysis was obtained by MetaPhlAn (version 4.0.1)[37] with default parameters based on metagenome reads. Community Beta

diversity was displayed by principal coordinate analysis (PCoA) based on the Bray–Curtis dissimilarity at the species level.

The neutral community model (NCM) was employed to assess the potential impact of stochastic processes on microbial community assembly[38]. This model is an adaptation of the neutral theory[39] tailored for extensive microbial populations. In general, the model posits that taxa abundant in the metacommunity are likely to be widespread since they have a higher chance of dispersing randomly among different sampling sites. Conversely, rare taxa are more prone to being lost across sites due to ecological drift, involving the stochastic loss and replacement of individuals. The parameter $R^2$ in this model serves as an indicator of the overall fit to the neutral model. The fitting statistics were calculated with 95% confidence intervals using bootstrapping with 1000 replicates. We conducted separate analyses using datasets from different ecosystems and regions.

To implement the NCM, the genera within each dataset were categorized into three partitions: those occurring more frequently than (above partition), less frequently than (below partition), or within (neutral partition) the 95% confidence interval of the NCM predictions. The R codes for NCM are available at https://github.com/Weidong-Chen-Microbial-Ecology/Stochastic-assembly-of-river-microeukaryotes.

## Microbial network construction

To discern variations in microbial community structures among ecosystems, microbial ecological networks (MENs) were constructed using ggClusterNet[40] (version 0.1.0) with parameters "$N = 0$, $r = 0.8$, $p = 0.05$, method = spearman". Instead of creating a single extensive network, we generated eight distinct networks, each corresponding to a specific ecosystem-region pair. Various network topological indices were calculated using ggClusterNet[40] (version 0.1.0) and listed in Supplementary Data 2.2, such as the number of edges (L), the number of vertices (n), connectance, mean clustering coefficient, number of clusters, centralization degree, centralization betweenness, centralization closeness, relative modularity (RM), and vulnerability (Vul). Particularly, the network vulnerability variation, used for stability assessments, was calculated through simulations of extinction, involving the random removal of 0% to 90% of nodes in the networks with 50 times repeats. The network RM, a measure of the extent to which a network is compartmentalized into different modules, was calculated as the ratio of the difference between the modularity of an empirical network and the mean of modularity from the networks over the mean of modularity from the random networks[41]. The detailed calculation of these network parameters has been fully described previously[40,42].

## Contig assembly and ORF prediction

Metagenome reads were assembled into contigs using Megahit[43] (version 1.1.3), with default parameters. Contigs with length ≥1000 bp were retained. Eukaryotic contigs were identified by cutting the contig into sub-contigs (windows = 1 kb, steps = 0.5 kb) and searching them against the NR (ftp://ftp.ncbi.nlm.nih.gov/blast/db) using DIAMOND (version 2.1.6)[44] BLASTX module. If 60% of the sub-contigs within one contig had the best hit as eukaryotic origin, the entire contig was recognized as eukaryotic origin, as previously described[6]. There were 4,071,847 (2.24%) contigs identified as eukaryotic origin. Specifically, 994,815 contigs were classified as Bacillariophyta, and 637,023 were classified as Chlorophyta. Gene open reading frames (ORFs) for the metagenomic assemblies were predicted using Prodigal (version 2.6.3)[45] with parameters "-p meta". Gene ORFs with length <100 bp were removed.

## Non-redundant gene catalog construction

A total of 522,671,245 gene ORFs were de-replicated by clustering at 80% aligned region with 95% nucleotide identity using MMseqs2 (version 13.45111)[46], with the parameters "easy-linclust -e 0.001, --min-seq-id 0.95", as used in the datasets of the TG2G and human gut

microbiome datasets[6,47], and "-c 0.80" was set to exclude the effects of the shorter genes[6]. Finally, a total of 296,289,678 non-redundant genes were obtained. The gene rarefaction analysis was performed by sampling the whole gene set to count the number of the gene clusters produced by MMseqs2, with a 5% sampling step 10 times.

## Gene catalog annotation and abundance

The non-redundant gene catalog was functionally annotated against the NR, UniRef50[48], Swiss-Prot[49], and databases using MMseqs2, with parameters "easy-search -e 0.01, --min-seq-id 0.3, --cov-mode 2, -c 0.8", as used in the TG2G dataset[6]. In addition, the de-replicated genes were also functionally annotated against the eggNOG Orthologous Groups database[27] (version 5.0) using eggNOG-mapper[50] (version 2.0.1), which integrates functional annotations from several sources, including the KEGG functional orthologs database[29], the carbohydrate-active enzymes (CAZy) database[51], and Cluster of Orthologous Groups categories (COG) database[52]. Moreover, antibiotic resistance genes (ARGs) were annotated by querying the de-replicated protein sequences against the Comprehensive Antibiotic Resistance Database (CARD, version 3.2.5)[53] using Resistance Gene Identifier (RGI, version 6.0.1)[54], with parameters "-t protein, -a DIAMOND, --include_loose, -d wgs". Furthermore, virulence factors were annotated by aligning gene sequences against the VFDB 2023[28] using DIAMOND blastp (version 2.1.6)[44] with an e-value threshold of $1 \times 10^{-5}$. The genes that failed to be mapped to any of these databases were designated as novel genes.

The abundance profile of the non-redundant gene catalog was produced using minimap2 (version 2.24)[55] and NGLess (version 1.5.0)[56] as previously described[33]: (1) Metagenome reads were mapped to the non-redundant gene catalog using minimap2; (2) Gene mapped reads were filtered with a 45-bp length and 95% identity cutoff; (3) The abundance was estimated as the number of short reads mapping to a given sequence, with multiple mappers (short reads mapping to more than one sequence) being distributed by unique mapper abundance (count function in NGLess with the parameter "multiple" set as "dist1"). For cross-sample comparisons, the abundance profiles were normalized by the library size.

## MAG construction and genome quality control

Metagenome-assembled genomes (MAG) were built and refined using MetaWRAP (version 1.3.0)[57] with default parameters, which integrated the binning results of metaBAT2 (version 2.12.1)[58], MaxBin2 (version 2.2.6)[59], and CONCOCT (version 1.0.0)[60]. The completeness, contamination, and strain heterogeneity were calculated using the 'lineage_wf' module of CheckM (version 1.2.2)[61] with default parameters.

A thorough taxonomy-based post-binning refinement process was then implemented for each MAG meeting the criteria of completeness ≥50% and contamination <10%. This process involved two key steps: 1) Removal of contigs identified as eukaryotic origin or viral origin. Eukaryotic contigs were identified as described above. Viral contigs were identified by VirSorter2 (version 2.2.4)[62], with parameters "--include-groups dsDNAphage, NCLDV, RNA, ssDNA, lavidaviridae, --min-length 5000, and --min-score 0.5"; and 2) Removal of contigs originating from a different species compared to the dominant organism present in the MAG, using MAGpurify (version 2.1.2)[63] with default modules and parameters ("phylo-markers", "clade-markers", "tetra-freq", "gc-content", and "known-contam"). This refinement process has removed a total of 246,813 contigs, constituting 1.8% of the initial 13,203,365 contigs within the bins.

CheckM (version 1.2.2)[61] was then re-conducted on these refined mags to calculate the completeness, contamination, and strain heterogeneity. The presence of rRNA and tRNA genes was identified using Infernal (version 1.1.3)[64] with models from the Rfam database[65] and the parameters "--cut_ga, --rfam". According to the standard of the MIMAG[23], only the refined MAGs meeting the medium and higher quality remained for the subsequent analysis, which was defined as

follows: 1) Medium-quality MAGs: completeness ≥ 50% and contamination <10%; and 2) High-quality MAGs: completeness > 90% and contamination <5% with the presence of the 23S, 16S, and 5S rRNA genes and at least 18 tRNAs. Ultimately, a total of 32,355 refined MAGs were retained, comprising 30,331 with medium quality and 2024 with high quality.

## MAGs clustering and taxonomy annotation

The 32,355 MAGs were clustered into 10,723 representative genome-based species using dRep (version 3.4.2)[66] based on >30% aligned fraction and a genome-wide ANI threshold of 95%, with parameters "-nc 0.3 and -sa 0.95", as used in the genomic catalog of Tibet glacier' microbiomes (TG2G)[6], Earth's microbiomes (GEM)[16], and human gut microbiomes[47]. The taxonomy annotation of the 10,723 species was performed using the module "classify_wf" of Genome Taxonomy Database Toolkit (GTDB-Tk, version 2.2.6)[67] against the GTDB release R214[24] with default parameters. The phylogenetic tree was generated through the maximum-likelihood placement of each MAG of this study in the GTDB-Tk reference tree using pplacer[68]. The subsequent visualization was achieved through ITOL (version 6)[69].

## Novelty of MAGs

The representative MAGs of the genome-based species were compared against 85,205 representative genomes from GTDB release R214[24], 746 representative MAGs of environmental origin from GEM[16], 3241 representative MAGs from TG2G[6], and 957 representative MAGs of the TARA ocean[25] using the "compare" module of dRep. The MAGs exhibiting an ANI < 0.95 and a coverage <0.3 compared to the reference genomes were designated as novel.

## Identification and clustering of BGCs

A total of 73,864 Biosynthetic gene clusters (BGCs) were predicted and identified on contigs ≥5 kb of the MAGs to reduce the risk of fragmentation[17], using AntiSMASH (version 6.1.1)[31], with parameters "--minlength 5000, --genefinding-tool prodigal-m". Each BGC was functionally annotated based on the predicted product types as defined by AntiSMASH and BiG-SCAPE, and categorized into eight groups using BiG-SCAPE (version 1.1.5)[70]: 'PKSI', 'PKSother', 'NRPS', 'RiPPs', 'Saccharides', 'Terpene', 'PKS-NRP_Hybrids', 'Others'. BGCs were then clustered into 18,414 gene cluster families (GCFs, 0.3 distance threshold) and 2681 gene cluster clans (GCCs, 0.7 distance threshold) using BiG-SCAPE[70] with default parameters. To prevent sampling biases in quantitative analysis (taxonomic and functional compositions of GCCs/GCFs, GCF and GCC distances to reference databases as well as GCF metagenomic abundances), the 73,864 BGCs were further dereplicated by retaining only the longest BGC per GCF per species[17], resulting in a total of 30,182 BGCs.

## Novelty of BGCs, GCFs, and GCCs

The novelty of BGCs, GCFs, and GCCs was determined by querying the 30,182 representative BGCs against a pre-processed BiG-FAM[71] reference database, using BiG-SLiCE (version 1.1.0)[32] with default parameters. This reference database contains 1,225,071 BGCs from 209,206 publicly available microbial genomes, which integrates MIBiG v2.0[72], RefSeq complete/draft bacteria, GenBank fungi, Genbank archaea, and other MAGs from different studies. The reference database has generated 29,955 GCF reference models. The rank first GCF of each query BGC with the calculated membership value ≤900 as used in BiG-FAM, was designated as the matched GCF. The query BGCs failed to map any of the reference GCFs were designated as novel. The query GCF with the proportion of novel BGCs <0.2 was designated as the novel, while the query GCC with that proportion <0.4 was designated as the novel. Finally, 18,549 out of the 30,182 BGCs, 10,128 out of 18,414 GCFs, and 1471 out of 2681 GCCs were designated as the novel.

## Abundance profiles of BGCs, GCFs, and GCCs

The abundance of each of the biosynthetic genes of a BGC was calculated using minimap2 (version 2.24)[55] and NGLess (version 1.5.0)[56], and then normalized by the library size as described in the gene-level profiling methods. The metagenomic abundance of a BGC was estimated as the median abundance of its biosynthetic genes (as defined by antiSMASH). The metagenomic abundance of each GCF and GCC was then computed as the sum of its representative BGCs.

## Reporting summary

Further information on research design is available in the Nature Portfolio Reporting Summary linked to this article.

## Data availability

The metagenomic sequencing data generated in this study have been deposited in the Genome Sequence Archive (GSA) section of the National Genomics Data Center under accession code CRA011511. The TPMC database is available at https://ngdc.cncb.ac.cn/tpmc, including all the metadata of the 498 metagenomic samples, TPMC non-redundant gene sequences and their annotations https://download.cncb.ac.cn/bigd/TPMC/TPMC_gene_catalog/, metagenome-assembled genomes and their annotations https://download.cncb.ac.cn/bigd/TPMC/TPMC_genome_catalog/, biosynthetic gene clusters and their annotations https://download.cncb.ac.cn/bigd/TPMC/TPMC_BGC/, as well as the non-redundant gene sequences of 607 samples across a 2,500-km transect in China https://download.cncb.ac.cn/bigd/TPMC/TLGC_gene_catalog. The referenced representative genomes including 85,205 from GTDB release R214[24], 22,746 of environmental origin from GEM[16], 3,241 from TG2G[6], 957 from the TARA ocean[25], are available at https://gtdb.ecogenomic.org/, https://portal.nersc.gov/GEM/genomes/, https://www.biosino.org/node/analysis/detail/OEZ008893, and https://merenlab.org/data/tara-oceans-mags/, respectively.

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

## Acknowledgements

This study was supported by the Second Tibetan Plateau Scientific Expedition and Research Program (STEP) (Grant No. 2019QZKK0304 to W.M.), the National Natural Science Foundation of China (Grant Nos. 32071465, 31871334, and 31671374 to K.N., Grant No. 32070418 to C.J., Grant No. U22A20454 to K.C.), the National Key Research and Development Program of China (Grant No. 2023YFA1800900 and 2018YFC0910502 to K.N.), the Science & Technology Fundamental Resources Investigation Program (Grant No. 2022FY100400 to C.J.). We thank the members of the Protist 10,000 Genomes Project (P10K) consortium for their helpful suggestions. Numerical computations were performed on the Wuhan Branch, Supercomputing Center, Chinese Academy of Sciences, China, and Hefei Advanced Computing Center. We thank Xiong Xiong, Tao Fang, Gaofei Song, and Xiuyun Cao of the Institute of Hydrobiology, Chinese Academy of Sciences for sharing environmental information data.

## Author contributions

M.C., J.X., W.M., and K.N. designed and supervised the study. M.C., S.L., P.Z., G.X., K.C., C.J., F.Y., H.H., P.Yang, G.L., Y.Z., S.B., and P.Yin participated in sample collection, investigations, and data analysis. M.C., S.L., and P.Z. led the data curation and processing. M.C. led the data analysis and visualization. M.C., S.L., P.Z., J.X. W.M., and K.N. led the results interpretation and writing of the manuscript. All authors contributed to the editing of the manuscript and approved the final version.

## Competing interests

The authors declare no competing interests.
