## [Peer Review File · Nature Communications]

A genome and gene catalog of the aquatic microbiomes of the Tibetan PlateauReviewer #1 (Remarks to the Author):

In this manuscript, Cheng and co-authors make public an exciting dataset from the Tibetan Plateau. This resource is fantastic and it will for sure be highly used and cited. The manuscript is well written, however, I bring forward several points that need to be addressed:

Line 33 – This should read. “The TPMC greatly increased genomic diversity by presenting 33,702 metagenome-assembled genomes that de-replicated into 11,437 representative genomospecies. For use of the term genomospecies see, for example <https://www.nature.com/articles/s41564-023-01435-6>

Please consider using this throughout

Line 41 – I do not see the reason to call TP a “microbial tower” in the abstract. Since most people only read the abstract, and the phrase is hard to grasp, it would be good to remove this analogy from the abstract.

Line 47 – The third pole requires either an explanation or to be removed.

Line 55 – Uncultivable is not quite right, see <https://www.nature.com/articles/s41579-020-00458-8> It is perhaps better to say uncultured, or uncultivated. I see you use the term a few more times, please reconsider them all.

Line 61 – cataloging microbial resources makes sense to me. But can we really protect microbial resources?

Line 132 – Do you refer to assembly size or estimated genome size? The actual genome size can't be known as you have incomplete genomes, but you could be referring either to assembly size of the MAGs or estimated genome size. Please see <https://www.frontiersin.org/articles/10.3389/fmicb.2021.761869/full>
Please also see your Table S3, because also there it is not clear in column H which one do you refer to.

Line 400 – how was the water collected?

Line 403 – was the desiccant directly added to the filter? What type of desiccant?

Line 404 – how long time occurred between sampling and storage at -80C

Line 412 to 420 – We need more details on the methods. Were the measurements done on filtered samples or whole samples?

Line 424 – How were the microorganisms washed from the membrane? It is possible that not all were washed and you lost some to membrane attachment.

Line 425 – What about cell lysis?

Line 568 – More studies have surfaced after GEMs, Tara and TG2G. A fairer comparison should also include all MAGs from GTDB since they have an extensive catalog that they continuously update.

Line 612 – Such article is only stronger because of the data it releases. Currently, CRA011511 is under embargo and can't be checked. It states that it will be released in June 2025. As a reviewer, I consider it necessary to make sure that the more than 400 metagenomes and the more than 11 000 representative MAGs are archived and findable.

Figure 1 panel b – are the PCoA based on annotated reads? Or MAGs abundance?

Figure 1 panel d – I could not find in the legend and description for this panel

Figure 2 panel c – it is not intuitive why you have two times novel OTUs, once time in red/orange and another time in blue. Please explain

Figure 2 panel d – this panel has 6 pointers with letters in them. What do they mean?

Figure 5 panel b and c. What is each dot? Pairwise comparison? Between what?

Figure 5 panel d. What is each dot here? One representative MAG in each sample, in all samples? Please explain

Figure 5 panel e and f. Also not clear what you are representing from the description.

Supplementary tables

- I am missing a supplementary table with all 498 metagenomes metadata including coordinates, and parameters measured as well as accession number.

- Table S3 with the MAGs is missing accession number per MAG.

Reviewer #2 (Remarks to the Author):

The manuscript titled "A Comprehensive Genome and Gene Catalog of Aquatic Microbiome in the Tibetan Plateau" explores microbial communities across six aquatic ecosystems using genome-centric metagenomics. The research underscores the vital role of the Tibetan Plateau (TP) in supplying water to a significant portion of Asia while highlighting the threats posed by global warming to its water body and indigenous microbial communities. The study involved the construction of a TP Microbial Catalog (TPMC) using 498 metagenomes from diverse water ecosystems.

Although the TPMC collection has the potential to reveal a substantial increase in genomic diversity, unveiling numerous novel species and gene clusters, I have major concerns regarding the methodology used to produce this collection of 33,000 MAGs and the oversight of important groups of organisms (Archaea, Eukaryotes), and viruses. Additionally, the study does not properly integrate current results with past findings from the same geographical region and studies from aquatic microbiomes across the world.

1. Recovering high-quality MAGs is not only essential for the conclusions of the current study but also for all future studies that would make use of this data. In this respect, I understand that the authors have used state-of-the-art binning software, made attempts to remove contigs of eukaryotic origin, and followed the MIMAG standards for defining MAG quality, keeping only those classified as high- and medium-quality. My major concern in this regard was the lack of a thorough taxonomy-based post-binning refinement step that would further reduce the risk of having chimeric bins. For this reason, I downloaded all the MAGs provided by the authors and chose only a subset of 712 MAGs for taxonomy checking due to time and resource limitations (all MAGs matching the pattern 2021-ZJZB*). Genes were predicted using Prodigal, and taxonomy annotation was performed for each protein by scanning with mmseqs against a database combining GTDB r214 proteomes and UniProt. The results showed that, in many cases, contigs of eukaryotic or purely viral origin were binned by the authors within MAGs classified as bacterial. Also, there is a frequent occurrence of contigs from different bacterial phyla mixed within the same MAG. A few examples that stand out include:

2021-ZJZB-W21_bin.103: Flavobacterium in the current study, with its longest contig (k127_4871380) of 135 kb derived from an eukaryote (Viridiplantae, Spirogyra).

2021-ZJZBSD-W51_bin.79: also as Flavobacterium, with contigs k127_397621 (12 kb) and k127_1752470 (5 kb) derived from eukaryotes (Ochrophyta).

2021-ZJZB-W32_bin.6: Cyanobacteria, containing contig k127_103457 (11 kb) derived from Bacteroidota (Arcicella).

2021-ZJZB-W2_bin.16: Actinobacteriota, containing contig k127_855421 (32 kb) derived from a Caudovirales phage.

There are numerous such occurrences only in the randomly chosen subset of 712 MAGs.

While CheckM is a useful tool for estimating completeness and contamination in Archaea and Bacteria, it is based on predefined collections of marker genes that do not take into account the presence of eukaryotic and viral sequences. For viral contig detection, the use of a dedicated software tool would be recommended. The authors' decision to include contigs as small as 1 kb increases the difficulty of removing spurious contigs even further. A threshold of 3-5 kb would significantly limit such issues.

2. Within the large collection of more than 33k MAGs recovered in this study, 135 MAGs were classified as Archaea, according to Supplementary Table S3. Information about this intriguing group of organisms should be provided in the main text.

3. Viruses are known to play key roles in microbial population dynamics and nutrient cycling within aquatic ecosystems. The present study completely ignores these entities. I recommend including a section about viruses recovered from metagenomes sequenced in this study.

4. The authors use a very limited number of external datasets (GEM, TG2G, TARA oceans) to compare and define the novelty of their recovered MAGs. Including more datasets, especially from similar environments, would strengthen the validity of findings presented in this study.

5. Some previous metagenomic studies from the Tibetan Plateau were not acknowledged (for example: <https://doi.org/10.1016/j.envres.2022.114847>,

<https://doi.org/10.1016/j.isci.2021.103439>). Overall, the authors make very few references to past studies from this same geographical region and do not properly integrate their own findings and observations with past work.

6. The abundance of the non-redundant gene catalog was calculated, however, there is no information provided regarding the most abundant organisms at the species level and no characterization of these organisms.

L440 "The taxonomic abundance profile of FMLG microbial community" - the FMLG community is not defined anywhere in the text.

Minor/Miscellaneous points:

1. The phylogenetic tree produced by GTDB-Tk with the classify_wf (Figure 3a) is not generated de novo but by inserting concatenated markers recovered from MAGs in this study in an existing phylogenetic tree with pplacer. The authors should make this clear in the methods and figure legend.

2. It would be preferable that individual classes be highlighted within Proteobacteria on the phylogenetic tree since this is a vast phylum.

3. The use of English throughout the manuscript, especially the Material and Methods section, as well as the title itself, requires improvements. Having a native English speaker check the manuscript could help improve the quality.

Reviewer #3 (Remarks to the Author):

The paper describes over 11,000 novel microbial species, ~60 million novel gene clusters and 35,000 novel biosynthetic gene clusters. 498 metagenomes from diverse Tibetan Plateau water ecosystems enabled comparative analysis of beta diversity, revealing negative correlations of similarity with distance and altitude, supporting the idea of a "microbial watertower", dispersing microbes throughout the region.

In my opinion, the work is significant to the field, with well-supported conclusions. While I did not find any major flaws, I request re-analysis with updated GTDB (r214 released April 2023) and better taxonomic consistency (see below). Minor comments below.

81: Remove ever

127: "The" instead of "he"

129/143: How many novel species had a MAG that passed quality score > 50 and also HQ?

146: How many OTUs were novel to both? How many are novel when compared to GTDB?

147: "In particular" instead of "Specifically"

152: How many mapped?

153: This conclusion could be improved by quantifying the contribution of TPMC. For instance, by reporting the increase in mapped % for all TP metagenomes when TPMC species representatives are added to the previous databases.

178: The new GTDB update (r214) includes 15 new phyla, etc. Can you repeat the analysis using the newest GTDBtk version?

202: How many unigenes were not annotated by any of the databases?

223: Please add a comment about why you think VF0082 is useful for pathogen monitoring, considering the prevalence of motility in non-pathogenic microbes.

224: "is" instead of "was"

274: Please also provide Pielou's evenness, as a better alpha diversity metric for between group comparisons.

278: Comment on why Tibet/Qilian saline lake samples cluster separately.

287: Refers to Figure 3d instead of Figure 2d

287: Remove "found mostly"

288: The "c" MAGs mentioned here appear to be located in an "other" phyla in Figure 3a, not in Proteobacteria? Are their bin names mentioned somewhere, so that I can find them in Table S3? This may be a NCBI vs GTDB taxonomy issue, if so, then please use the same taxonomy throughout (e.g. switch completely to GTDB by referring to these "other" MAGs as Myxococota or revert to NCBI by changing Bdellovibrionota to Proteobacteria, etc.).

301: "adaptation" instead of "adaption"

325: "representing one of" instead of "representing the one of"

332: "biogeography" typo

342: Sentence is a bit clunky, please reword.

357: "and store": Is this referring to frozen sample storage? It suggests cultured organisms.

376: "Everything is everywhere" does not suggest that we have isolated genomes for everything. I don't think there is a contradiction here.

386: "advancing" sounds too direct. You are providing data that may contribute to these areas, not directly advancing them.

Methods: please specify how the "similarity" was calculated for Figure 5.

Table S3: please report GTDBtk notes and warnings columns

Figure S7: Why is the Qilian glacier boxplot orange in the first plot, but green in the others?

Response letter

Reviewer #1

In this manuscript, Cheng and co-authors make public an exciting dataset from the Tibetan Plateau. This resource is fantastic and it will for sure be highly used and cited. The manuscript is well written, however, I bring forward several points that need to be addressed:

1. Line 33 – This should read. “The TPMC greatly increased genomic diversity by presenting 33,702 metagenome-assembled genomes that de-replicated into 11,437 representative genomespecies.

For use of the term genomespecies see, for example <https://www.nature.com/articles/s41564-023-01435-6>. Please consider using this throughout.

Answer: Many thanks for your comments. In response to the comments, we have used the term “genomespecies” throughout the revised manuscript. An example of the revision has been listed below (**Lines 28 to 30**). We have highlighted all the instances of the term “genomespecies” throughout the revised manuscript for your review.

[Main text]

Lines 28 to 33: The TPMC greatly increased genomic diversity by presenting 32,355 metagenome-assembled genomes that de-replicated into 10,723 representative genomespecies, while 88% of these genomespecies were unannotated.

2. Line 41 – I do not see the reason to call TP a “microbial tower” in the abstract. Since most people only read the abstract, and the phrase is hard to grasp, it would be good to remove this analogy from the abstract.

Answer: Many thanks for your comments. In response to the comments, we have removed this analogy from the abstract to avoid ambiguity (**Lines 35 to 37**).

[Main text]

Lines 35 to 37: As distance and altitude differed from the TP increased, we observed a decline in microbial compositional similarity and the shared gene count with the TP microbiome, implying a dispersal pattern in biogeography.

3. Line 47 – The third pole requires either an explanation or to be removed.

Answer: Many thanks for your comments. In response to the comments, we have added an explanation and references to the term “third pole” (**Lines 43 to 45**).

[Main text]

Lines 43 to 45: The Tibetan Plateau (TP), spanning over 2.5 million square kilometers and averaging an altitude of over 4,500 meters, is the highest and largest plateau globally, with its ice fields containing the largest reserve of freshwater outside the polar regions, thus known as the "third pole"¹⁻³.

4. Line 55 – Uncultivable is not quite right, see <https://www.nature.com/articles/s41579-020-00458-8> It is perhaps better to say uncultured, or uncultivated. I see you use the term a few more times, please reconsider them all.

Answer: Many thanks for your comments. In response to the comments, we have changed the word to “uncultivated” throughout the revised manuscript and highlighted them in the revised manuscript for your review (**Lines 52 to 54** and **Lines 63 to 65**).

[Main text]

Lines 52 to 54: The diverse water ecosystems within the TP, encompassing glaciers, wetlands, hot springs, lakes, and rivers, harbor a rich diversity of aquatic microorganisms⁴⁻⁹, including those from uncultivated new lineages⁴, and exhibit substantial biosynthetic potential⁴.

Lines 63 to 65: In recent years, advancements in sequencing technologies and computational methods within metagenomics have facilitated the reconstruction of genomes from previously uncultivated microorganisms.

5. Line 61 – cataloging microbial resources makes sense to me. But can we really protect microbial resources?

Answer: Many thanks for your comments. We agree with you that protecting microbial resources poses greater challenges than cataloging them. The impact of irreversible environmental changes and human activities has already affected microbial resources, a trend likely to persist. Therefore, preserving the current microbial diversity becomes imperative to prevent further diminishment. Cataloging microbial resources could at least play a role in mining, understanding, and storing such diversity, telling people what we have and what we will lose. Furthermore, the catalog, also serving as an educational resource, raises public awareness about the importance of microbial diversity, fostering support for conservation initiatives. Its global nature encourages international collaboration, leading to the development of common standards and policies for the responsible use and protection of microbial resources. In essence, a well-maintained microbial catalog is instrumental in advancing scientific research, guiding ethical practices, and ensuring the sustainable use of microbial resources on a global scale. We acknowledge that there is still a long way to go in truly protecting microbial resources, and cataloging would be the first step toward this target.

6. Line 132 – Do you refer to assembly size or estimated genome size? The actual genome size can't be known as you have incomplete genomes, but you could be referring either to assembly size of the MAGs

or estimated genome size. Please see

<https://www.frontiersin.org/articles/10.3389/fmicb.2021.761869/full>

Please also see your Table S3, because also there it is not clear in column H which one do you refer to.

Answer: Many thanks for your comments. We have referred the size to the assembly size of the MAGs in the revised manuscript. We have replaced “genome size” with “assembly size” throughout the manuscript and supplementary tables. Furthermore, we have highlighted all the instances of the term “assembly size” in the revised manuscript for your review (**Lines 152, 156, and 942**).

[Main text]

Line 152: The assembly sizes of these MAGs showed no significant difference...

Line 156: ...wetland microbiomes (2.6 Mb, 1.7–3.2Mb) showed the highest assembly sizes...

Line 942: the assembly size of the encompassed MAGs with the largest count of BGCs.

[Table S3]

Table S3 Characteristics of the metagenome-assembled genomes in the Tibetan Plateau aquatic genome and gene catalog.

Region	Num_contigs	N50	Assembly_Size	GC	Completeness	Contamination
Tibet	519	6150	2541731	0.33878	79.38	3.52
Tibet	107	49097	3165745	0.62115	84.21	1.89
Tibet	529	3345	1640940	0.43342	79.13	2.08
Tibet	19	119586	1171239	0.38222	82.02	1.12
Tibet	591	8531	3766984	0.37543	76.67	2.6
Tibet	980	3502	3113623	0.6364	60.14	3.51

Due to the large size of this table, only part of it is displayed here. Please see the full table in Supplementary tables.xlsx.

7. Line 400 – how was the water collected?

Answer: Many thanks for your comments. A water sampler was used for collection. We have added the related information to the revised manuscript (**Lines 442 to 444**).

[Main text]

Lines 442 to 444: For sampling, 5 L of surface water (10–30 cm depth) was collected at each station using a water sampler. The collected water was filtered using a 200 µm filter to remove insoluble impurities, large organisms, aquatic plants, and other debris.

8. Line 403 – was the desiccant directly added to the filter? What type of desiccant?

Answer: Many thanks for your comments. The membrane was immediately placed in aluminum foil and

kept in vials with Silicagel desiccant. We have added this information to the revised manuscript (**Lines 445 to 448**).

[Main text]

Lines 445 to 448: The membrane was immediately placed in aluminum foil, deposited in vials containing Silica gel desiccant, subjected to desiccation, and stored at -80°C in the laboratory for 3–7 days before subsequent metagenomic sequencing.

9. Line 404 – how long time occurred between sampling and storage at -80°C

Answer: Many thanks for your comments. Sequencing was usually performed after completing sampling in an entire area. Therefore, the storage duration of membrane samples in a -80°C freezer is typically 3–7 days. We have added this information to the revised manuscript (**Lines 445 to 448**).

[Main text]

Lines 445 to 448: The membrane was immediately placed in aluminum foil, deposited in vials containing Silica gel desiccant, subjected to desiccation, and stored at -80°C in the laboratory for 3–7 days before subsequent metagenomic sequencing.

10. Line 412 to 420 – We need more details on the methods. Were the measurements done on filtered samples or whole samples?

Answer: Many thanks for your comments. The measurements for environmental factors of water samples were conducted on whole water samples. We have added this information to the revised manuscript (**Line 448**).

[Main text]

Line 448: Environmental parameters of the whole water samples were measured at each station.

11. Line 424 – How were the microorganisms washed from the membrane? It is possible that not all were washed and you lost some to membrane attachment.

Answer: Many thanks for your comments. To ensure thorough elution of microorganisms from the membrane, we first cut the membrane into small pieces with scissors (cut more than eight times). Subsequently, the cut membrane was placed in a 1.5 mL EP tube, and 800 μL of CTAB extraction buffer (2% CTAB, 100mmol/L Tris-HCl (pH 8.0), 20 mmol/L EDTA, 1.4mol/L NaCl) were added for mixing. Then, grinding beads were added to it, and after ground using a grinder Tissuelyser-192L (Jingxin, Shanghai, China, 55 Hz, 5 min), it was heated at 70 degrees Celsius for 10 minutes in a water bath to ensure sufficient DNA extraction. We have added these details to the methods section for clarity (**Lines 467 to 472**).

[Main text]

Lines 467 to 472: First, the membrane was cut into small pieces using scissors (cut more than eight times). Subsequently, the cut membrane was placed in a 1.5 mL EP tube, and 800 μ L of CTAB extraction buffer (2% CTAB, 100 mmol/L Tris-HCl (pH 8.0), 20 mmol/L EDTA, 1.4 mol/L NaCl) was added for mixing. Then, grinding beads were added to it, and after ground using a grinder Tissuelyser-192L (Jingxin, Shanghai, China, 55 Hz, 5 min), it was heated at 70 degrees Celsius for 10 minutes in a water bath to ensure sufficient DNA extraction.

12. Line 425 – What about cell lysis?

Answer: Many thanks for your comments. We employed 2% CTAB extraction buffer to lyse the cells, and this step was conducted in conjunction with the rinsing of microorganisms on the membrane and DNA extraction (**Lines 468 to 470**).

[Main text]

Lines 468 to 470: Subsequently, the cut membrane was placed in a 1.5 mL EP tube, and 800 μ L of CTAB extraction buffer (2% CTAB, 100 mmol/L Tris-HCl (pH 8.0), 20 mmol/L EDTA, 1.4 mol/L NaCl) was added for mixing.

Line 568 – More studies have surfaced after GEMs, Tara and TG2G. A fairer comparison should also include all MAGs from GTDB since they have an extensive catalog that they continuously update.

Answer: Many thanks for your comments. We have recruited 85,205 representative genomes from GTDB R214 database for comparison and found 894 (8.4%) TPMC genomespecies could map to them. We have fully updated these results in the **main text (Lines 169 to 172)**, **Figure 2c**, **Figure 3a**, **Figure S1**, and **Table S5**.

[Main text]

Lines 169 to 172: The 10,723 genomespecies demonstrated substantial novelty, with 91.64%, 98.46%, 98.78%, and 99.98% exhibiting low sequence identity to 85,205 GTDB genomes, 22,746 GEM genomes, 3,241 TG2G genomes, and 957 TARA genomes, respectively (**Supplementary Table 5**).

[Figure 2c]

Fig. 2 | Environmental and geographical distribution of metagenome-assembled genomes and their species-level clustering. c, TPMC MAGs are clustered based on 95% ANI and 30% AF, resulting in 10,723 genomespecies. Representative genomes from GTDB R214 (n = 85,205), GEM (n = 22,746), TG2G (n = 3,241), and TARA (n = 957) were then included in the clustering to identify the novelty of the TPMC genomespecies.

[Figure 3a]

Only part of Figure 3a is displayed here for review.

Fig. 3 | Novelty and phylogenomic distribution of the metagenome-assembled genomes and their biosynthetic potential. a, A phylogenetic tree built for all the TPMC genomespecies (n = 10,723) based on a concatenated alignment of 120 universally distributed bacterial single-copy genes and placement of each MAG in the GTDB-Tk reference tree. Two classes of the phylum Pseudomonadota are marked in the tree. The color of the branches indicates the phylum of the genomespecies. The outer layers indicate, for each genomespecies, the region and water ecosystem labels based on its MAGs, the count of MAGs, the largest count of BGCs among its MAGs, and the novelty (compared to the representative genomes of GTDB, GEM, and TG2G catalogs, respectively).

[Figure S1]

Only part of Figure S1 is displayed here for review.

Fig. S1 | Overview of the metagenomic analysis.

[Table S5]

Table S5 The comparison between the representative TPMC MAGs to the representative genomes from GTDB, GEM, TG2G, and TARA catalogs.

10,723 representative TPMC MAGs compared with 85,205 representative genomes of GTDB R214 (896 (8.4%) matched).						
TPMC representative MAG	Matched reference MAG	ANI	Coverage	TPMC OTU	Taxonomic annotation	TPMC ecosystem type
2020_DTH4-W1_bin.51_purify	GCA_003671255.1_genomic	0.967328	0.981404959	3786_3	d__Bacteria;p__Planctomycetota;c__Planctomycetia;o__Gemmatales;f__Gemmataceae;g__UBA969;s__UBA969 sp003671255	River
2019_bqzb-4_bin.52_purify	GCA_021738605.1_genomic	0.970294	0.977232925	2450_1	d__Bacteria;p__Bacteroidota;c__Bacteroidia;o__Chitinophagales;f__Chitinophagaceae;g__SXYR01;s__SXYR01 sp021738605	River
2020_CRC-20W-1-14m-1_bin.48_purify	GCA_945872875.1_genomic	0.987787	0.967346939	2301_1	d__Bacteria;p__Bacteroidota;c__Bacteroidia;o__Flavobacteriales;f__Flavobacteriaceae;g__Flavobacterium;s__Flavobacterium psychrophilum_N	Freshwater lake
2020_KMC-20W-1-20m-1_bin.204_purify	GCF_017355915.2_genomic	0.999833	0.965401249	3892_1	d__Bacteria;p__Bacteroidota;c__Bacteroidia;o__Cytophagales;f__Spirosomaceae;g__Tellurbacter;s__Tellurbacter sp017355915	Freshwater lake
H4-1_bin.20_purify	GCF_000828655.1_genomic	0.985647	0.964232489	4293_1	d__Bacteria;p__Thermotogota;c__Thermotogae;o__Thermotogales;f__DSM-5069;g__Pseudothermotoga_A;s__Pseudothermotoga_A caldfontis	Hot spring
2021-CBL-W3_bin.99_purify	GCA_009703465.1_genomic	0.961693	0.958715596	2851_2	d__Bacteria;p__Actinomycetota;c__Actinomycetia;o__Actinomycetales;f__Microbacteriaceae;g__Rhodoluna;s__Rhodoluna sp009703465	Wetland
2021-YM02-W1_bin.1_purify	GCA_018402665.1_genomic	0.962063	0.955974843	2017_1	d__Bacteria;p__Bacteroidota;c__Bacteroidia;o__Flavobacteriales;f__Schleiferiaceae;g__UBA10364;s__UBA10364 sp018402665	Saline lake
2020_KMC-20W-1-20m-1_bin.8_purify	GCA_903827085.1_genomic	0.97524	0.95505618	4884_1	d__Bacteria;p__Actinomycetota;c__Acidimicrobia;o__Acidimicrobiales;f__UBA8139;g__F1-20-MAGs160;s__F1-20-MAGs160 sp903827085	Freshwater lake

Due to the large size of this table, only part of it is displayed here. Please see the full table in the Supplementary tables.xlsx.

13. Line 612 – Such article is only stronger because of the data it releases. Currently, CRA011511 is under embargo and can't be checked. It states that it will be released in June 2025. As a reviewer, I consider it necessary to make sure that the more than 400 metagenomes and the more than 11 000 representative MAGs are archived and findable.

Answer: Many thanks for your comments. The temporal link provided for review has expired during the review process. We have rectified this issue by updating the link in the revised manuscript (Lines 653 to 655). Upon publication of the paper, the associated data will be made public. During the review process, you can view the metagenome data through the temporal link: <https://ngdc.cnpc.ac.cn/gsa/s/ZHbsW74l>

[Main text]

Lines 653 to 655: Metagenomic sequencing data of TPMC are available in the Genome Sequence Archive (GSA) section of the National Genomics Data Center (project accession number CRA011511, <https://ngdc.cncb.ac.cn/gsa/s/ZHbsW74l>)

[GSA page]

A part of the GSA page is displayed here for review.

CRX725440	2021-HHSDK-W2	metagenome	MGISEQ-2000	PRJCA017393	SAMC2804959	2025-06-20	Checked OK Confidential
Run Accession	Title / Alias	Run Data File Information				Release Date	Run Status
CRR807581	2021-HHSDK-W2	File: 2021-HHSDK-W2_1.fastq.gz File: 2021-HHSDK-W2_2.fastq.gz	Status: Processed Succeed Status: Processed Succeed			2025-06-20	Processed Succeed Confidential
CRX725439	2021-HHSDK-W1	metagenome	MGISEQ-2000	PRJCA017393	SAMC2804958	2025-06-20	Checked OK Confidential
Run Accession	Title / Alias	Run Data File Information				Release Date	Run Status
CRR807580	2021-HHSDK-W1	File: 2021-HHSDK-W1_1.fastq.gz File: 2021-HHSDK-W1_2.fastq.gz	Status: Processed Succeed Status: Processed Succeed			2025-06-20	Processed Succeed Confidential
CRX725438	2021-HHSD-W3	metagenome	MGISEQ-2000	PRJCA017393	SAMC2804957	2025-06-20	Checked OK Confidential
Run Accession	Title / Alias	Run Data File Information				Release Date	Run Status
CRR807579	2021-HHSD-W3	File: 2021-HHSD-W3_1.fastq.gz File: 2021-HHSD-W3_2.fastq.gz	Status: Processed Succeed Status: Processed Succeed			2025-06-20	Processed Succeed Confidential

14. Figure 1 panel b – are the PCoA based on annotated reads? Or MAGs abundance?

Answer: Many thanks for your comments. The PCoA are based on the genera profiles produced by the MetaPhlAn (version 4.0.1)¹⁰. We have added this information in the revised manuscript to avoid ambiguity (Lines 895 to 896).

[Main text]

Lines 895 to 896 | Figure 1b: Microbial community compositions plotted on a PCoA plot based on Bray–Curtis distance of the genera profiles produced by the MetaPhlAn (version 4.0.1)¹⁰.

15. Figure 1 panel d – I could not find in the legend and description for this panel

Answer: Many thanks for your comments. We have missed the symbol “d” in the legend. We have fixed this in the revised manuscript (Lines 902 to 903).

[Main text]

Lines 893 to 894 | Figure 1d: d, The MENs of five water ecosystems in Tibet were constructed based on Spearman correlations of genera relative abundances produced by MetaPhlAn (version 4.0.1)¹⁰.

16. Figure 2 panel c – it is not intuitive why you have two times novel OTUs, once time in red/orange and another time in blue. Please explain

Answer: Many thanks for your comments. It is due to that we have compared the representative TPMC OTUs to representative GEM OTUs and representative TG2G OTUs, respectively. GEM shares many types of water ecosystems but different geography positions, while TG2G currently represents another

genomic catalog of the Tibetan Plateau microbiome. Therefore, we have displayed two times novel OTUs corresponding to the different reference databases. These methods could be referred to the Table 1 in the TG2G paper: <https://doi.org/10.1038/s41587-022-01367-2>. Moreover, we have additionally recruited the 85,205 representative genomes of GTDB (R214) for comparisons, as you have recommended in the previous comments. Furthermore, we have reported the number of MAGs that did not map to any of the four databases (GTDB R214, GEM, TG2G, TARA) in the revised manuscript (**Lines 172 to 174**).

[Main text]

Lines 172 to 174: A total of 9,698 genomospecies did not map to any of these catalog genomes, of which 6,335 passed a quality score >50, and 465 met high-quality criteria.

17. Figure 2 panel d – this panel has 6 pointers with letters in them. What do they mean?

Answer: Many thanks for your comments. These letters denote the genomospecies that contained MAGs recovered solely from a specific ecosystem. We have added this information to the legend for clarity (**Lines 919 to 920**).

[Main text]

Lines 919 to 920 | Figure 2d: The genomospecies that contained MAGs recovered solely from a specific ecosystem were annotated by letters.

18. Figure 5 panel b and c. What is each dot? Pairwise comparison? Between what?

Answer: Many thanks for your comments. Each dot represents the pairwise comparisons in similarity quantified by the Bray–Curtis distance of species profiles produced by MetaPhlAn (version 4.0.1)¹⁰, against geographical distance (**Figure 5b**), and altitude (**Figure 5c**), which were conducted on samples within the TP (first ladder step), as well as between the TP and the other two ladder steps. We have added this information to the revised manuscript for clarity (**Lines 951 to 953 and Lines 958 to 962**).

[Main text]

Lines 951 to 953 | Figure 5 b and c: The microbial species profiles become less similar between samples in the same water ecosystem, as the distance (b) and altitude (c) difference increase.

Lines 958 to 962 | Figure 5: In all scatter plots, each dot represents a pairwise comparison in 1) microbial similarity quantified by the Bray–Curtis distance of species profiles produced by MetaPhlAn (version 4.0.1)¹⁰; 2) shared gene count; 3) geographical distance; and 4) altitude, which were conducted on samples within the TP (first ladder step), as well as between the TP and the other two ladder steps.

19. Figure 5 panel d. What is each dot here? One representative MAG in each sample, in all samples? Please explain

Answer: Many thanks for your comments. Each dot represents the relative abundance (\log_{10}) of the

phylum of each sample, produced by the MetaPhlAn (version 4.0.1). We have added this information to the revised manuscript for clarity (**Lines 953 to 956**).

[Figure legends]

Lines 953 to 956 | Figure 5d: The boxplots show the abundance variation of two phyla in the river samples across three ladders. Each dot represents the relative abundance (\log_{10}) of the phylum of each sample, produced by the MetaPhlAn (version 4.0.1)¹⁰. **** $P < 0.0001$; ns, no significance; Mann-Whitney-Wilcoxon test.

20. Figure 5 panel e and f. Also not clear what you are representing from the description.

Answer: Many thanks for your comments. Each dot represents the pairwise comparisons in shared gene count, against geographical distance (**Figure 5e**) or altitude (**Figure 5f**) conducted on samples within the TP (first ladder step), as well as between the TP and the other two ladder steps. We have added this information to the Figure legend for clarity (**Lines 956 to 957 and Lines 944 to 945**).

[Main text]

Lines 956 to 957 | Figure 5e and f: The number of the shared genes decreases between samples in the same water ecosystem, as the distance (e) and altitude (f) difference increase.

Lines 958 to 962 | Figure 5: In all scatter plots, each dot represents a pairwise comparison in 1) microbial similarity quantified by the Bray–Curtis distance of species profiles produced by MetaPhlAn (version 4.0.1)¹⁰; 2) shared gene count; 3) geographical distance; and 4) altitude, which were conducted on samples within the TP (first ladder step), as well as between the TP and the other two ladder steps.

21. Supplementary tables

- I am missing a supplementary table with all 498 metagenomes metadata including coordinates, and parameters measured as well as accession number.

- Table S3 with the MAGs is missing accession number per MAG.

Answer: Many thanks for your comments. All the 498-metagenome metadata including coordinates, and the measured parameters, as well as the accession numbers, have been deposited in a table in the TPMC database <https://ngdc.cnecb.ac.cn/tpmc>. For clarity, we have added the information on the metadata availability to the section “Data availability”, according to your comments (**Lines 655 to 656**).

Moreover, all the MAGs were packaged and deposited in the TPMC database as well, and could be downloaded through the link: https://download.cnecb.ac.cn/bigd/TPMC/TPMC_genome_catalog/, thus having no accession number (**Lines 658 to 660**).

[Main text]

Lines 653 to 663: Metagenomic sequencing data of TPMC are available in the Genome Sequence

Archive (GSA) section of the National Genomics Data Center (project accession number CRA011511, <https://ngdc.cnbc.ac.cn/gsa/s/ZHbsW74l>). The TPMC database is available at <https://ngdc.cnbc.ac.cn/tpmc>, including all the metadata of the 498 metagenomic samples, TPMC non-redundant gene sequences and their annotations (https://download.cnbc.ac.cn/bigd/TPMC/TPMC_gene_catalog/), metagenome-assembled genomes and their annotations (https://download.cnbc.ac.cn/bigd/TPMC/TPMC_genome_catalog/), biosynthetic gene clusters and their annotations (https://download.cnbc.ac.cn/bigd/TPMC/TPMC_BGC/), as well as the TLGC non-redundant gene sequences (https://download.cnbc.ac.cn/bigd/TPMC/TLGC_gene_catalog/).

[Metadata in the database]

Sample ID	Metagenome accession number	Location	Ecoregions	Region	Biome	Latitude	Longitude	Altitude	Chlorophyll (RFU)	Chlorophyll (ug/L)	Depth (m)	Fluorescent dissolved organic matter (IDOM, QSU)	Fluorescent dissolved organic matter (IDOM, RFU)
2020_NRPC-20W-1-2.5m-1	CRR807380	Nagqu	Tibetan Plateau alpine shrublands and meadows	Tibet	Saline lake	31.309908	91.447875	4524	1.23	4.91	2.384	11.22	3.74
2020_NRPC-20W-1-7.5m-1	CRR807381	Nagqu	Tibetan Plateau alpine shrublands and meadows	Tibet	Saline lake	31.309908	91.447875	4524	1.01	4.03	4.517	11.25	3.75
2020_NRPC-20W-1-10m-1	CRR807382	Nagqu	Tibetan Plateau alpine shrublands and meadows	Tibet	Saline lake	31.309908	91.447875	4524	0.83	3.32	7.057	11.41	3.8

Due to the large size of this table, only part of it is displayed here. Please see the full table at the <https://ngdc.cnbc.ac.cn/tpmc>

Reviewer #2

The manuscript titled "A Comprehensive Genome and Gene Catalog of Aquatic Microbiome in the Tibetan Plateau" explores microbial communities across six aquatic ecosystems using genome-centric metagenomics. The research underscores the vital role of the Tibetan Plateau (TP) in supplying water to a significant portion of Asia while highlighting the threats posed by global warming to its water body and indigenous microbial communities. The study involved the construction of a TP Microbial Catalog (TPMC) using 498 metagenomes from diverse water ecosystems.

Although the TPMC collection has the potential to reveal a substantial increase in genomic diversity, unveiling numerous novel species and gene clusters, I have major concerns regarding the methodology used to produce this collection of 33,000 MAGs and the oversight of important groups of organisms (Archaea, Eukaryotes), and viruses. Additionally, the study does not properly integrate current results with past findings from the same geographical region and studies from aquatic microbiomes across the world.

1. Recovering high-quality MAGs is not only essential for the conclusions of the current study but also for all future studies that would make use of this data. In this respect, I understand that the authors have used state-of-the-art binning software, made attempts to remove contigs of eukaryotic origin, and followed the MIMAG standards for defining MAG quality, keeping only those classified as high- and medium-quality. My major concern in this regard was the lack of a thorough taxonomy-based post-binning refinement step that would further reduce the risk of having chimeric bins. For this reason, I downloaded all the MAGs provided by the authors and chose only a subset of 712 MAGs for taxonomy checking due to time and resource limitations (all MAGs matching the pattern 2021-ZJZB*). Genes were predicted using Prodigal, and taxonomy annotation was performed for each protein by scanning with mmseqs against a database combining GTDB r214 proteomes and UniProt. The results showed that, in many cases, contigs of eukaryotic or purely viral origin were binned by the authors within MAGs classified as bacterial. Also, there is a frequent occurrence of contigs from different bacterial phyla mixed within the same MAG. A few examples that stand out include:

2021-ZJZB-W21_bin.103: Flavobacterium in the current study, with its longest contig (k127_4871380) of 135 kb derived from an eukaryote (Viridiplantae, Spirogyra).

2021-ZJZBSD-W51_bin.79: also as Flavobacterium, with contigs k127_397621 (12 kb) and k127_1752470 (5 kb) derived from eukaryotes (Ochrophyta).

2021-ZJZB-W32_bin.6: Cyanobacteria, containing contig k127_103457 (11 kb) derived from Bacteroidota (Arcicella).

2021-ZJZB-W2_bin.16: Actinobacteriota, containing contig k127_855421 (32 kb) derived from a Caudovirales phage.

There are numerous such occurrences only in the randomly chosen subset of 712 MAGs.

While CheckM is a useful tool for estimating completeness and contamination in Archaea and Bacteria, it is based on predefined collections of marker genes that do not take into account the presence of

eukaryotic and viral sequences. For viral contig detection, the use of a dedicated software tool would be recommended. The authors' decision to include contigs as small as 1 kb increases the difficulty of removing spurious contigs even further. A threshold of 3-5 kb would significantly limit such issues.

Answer: Many thanks for your comments. According to your suggestions, we have conducted a thorough taxonomy-based post-binning refinement process, encompassing three steps (**Lines 577 to 586**):

1) Removal of Contigs of Eukaryotic Origin:

- Taxonomic annotation of contigs involved cutting them into sub-contigs (windows=1kb, steps=0.5kb)
- Subsequently, these sub-contigs were searched against the NR using DIAMOND (version 2.1.6)¹¹ BLASTX module.
- Contigs were identified as eukaryotic if 60% of their sub-contigs had the best hit as eukaryotic origin, aligning with established methods⁴.

2) Removal of Viral-Origin Contigs:

- Leveraging your suggestion, we employed VirSorter2 (version 2.2.4)¹² to discern viral contigs, with parameters “--include-groups dsDNAphage, NCLDV, RNA, ssDNA, laavidaviridae, --min-length 5000, and --min-score 0.5”.

3) Decontamination of Contigs from Different Species:

- MAGpurify (version 2.1.2)¹³ facilitated the decontamination process, employing default modules and parameters: phylo-markers, clade-markers, tetra-freq, gc-content, and known-contam.

This three-step refinement process effectively eliminated 246,813 contigs, comprising 1.87% of the initial 13,233,761 contigs within the bins. Subsequently, we verified that the examples you highlighted were indeed removed during this process.

Furthermore, we have re-conducted the following analysis based on the refined MAGs:

1) Quality check (Lines 588 to 596):

- Re-executed checkM (version 1.2.2)¹⁴ and Infernal(version 1.1.3)¹⁵ to assess the quality of the refined MAGs, resulting in 32,355 retained MAGs, including 30,331 of medium quality and 2,024 of high quality, according to MIMAG¹⁶.

2) Taxonomic annotation (Lines 599 to 607):

- Re-ran the dRep (version 3.4.2)¹⁷ and GTDB-Tk (version 2.2.6)¹⁸ with up-to-date GTDB release r214 to obtain the taxonomic annotations. The 32,355 MAGs were clustered into 10,723 representative species-level OTUs.

3) Figure and Table Reproduction:

- Re-produced **Figure 2, Figure 3, Figure S4**, as well as **Tables S3 to S5**, considering the effects resulting from the removal of MAGs failing to meet the MIMAG medium quality

level post-refinement.

All relevant results and text in the revised manuscript have been thoroughly updated based on the 32,355 refined MAGs (e.g., **Lines 145 to 152**). Notably, the refinement process resulted in a 4% decrease in the number of MAGs, a 1.7% decrease in the number of BGCs, and a 0.6% decrease in mean contamination. Moreover, the utilization of the latest GTDB database (R214) resulted in a 34% decrease in the proportion of unclassified families, a 20% decrease in unclassified genera, and an 8.7% decrease in unclassified species. The taxonomy-based refinement has markedly enhanced the accuracy and applicability of these MAG resources for fellow researchers.

[Main text]

Lines 145 to 152: We performed metagenomic assembly and binning on the 498 metagenomes and recovered a total of 32,355 MAGs. These MAGs met or exceeded the medium-quality criteria outlined in the Minimum Information about a Metagenome-Assembled Genome (MIMAG) standard¹⁶, with mean completeness of 78.1% and mean contamination of 2.6%. Of these MAGs, 25,017 exhibited a quality score above 50 (defined as completeness – (5 × contamination)), and 2,024 were assigned as high quality, featuring the presence of the 23S, 16S, and 5S rRNA genes and at least 18 tRNAs (**Fig. 2a–b** and **Supplementary Table 3**).

Lines 577 to 586: A thorough taxonomy-based post-binning refinement process was then implemented for each MAG meeting the criteria of completeness $\geq 50\%$ and contamination $< 10\%$. This process involved two key steps: 1) Removal of contigs identified as eukaryotic origin or viral origin. Eukaryotic contigs were identified as described above. Viral contigs were identified by VirSorter2 (version 2.2.4)¹², with parameters “--include-groups dsDNAphage, NCLDV, RNA, ssDNA, lavidaviridae, --min-length 5000, and--min-score 0.5”; and 2) Removal of contigs originating from a different species compared to the dominant organism present in the MAG, using MAGpurify (version 2.1.2)¹³ with default modules and parameters (“phylo-markers”, “clade-markers”, “tetra-freq”, “gc-content”, and “known-contam”). This refinement process has removed a total of 246,813 contigs, constituting 1.8% of the initial 13,203,365 contigs within the bins.

Lines 588 to 596: CheckM (version 1.2.2)¹⁴ was then re-conducted on these refined mags to calculate the completeness, contamination, and strain heterogeneity. The presence of rRNA and tRNA genes was identified using Infernal (version 1.1.3)¹⁵ with models from the Rfam database¹⁹ and the parameters “--cut_ga, --rfam”. According to the standard of the MIMAG¹⁶, only the refined MAGs meeting the medium and higher quality remained for the subsequent analysis, which was defined as follows: 1) Medium-quality MAGs: completeness $\geq 50\%$ and contamination $< 10\%$; and 2) High-quality MAGs: completeness $> 90\%$ and contamination $< 5\%$ with the presence of the 23S, 16S, and 5S rRNA genes and at least 18 tRNAs. Ultimately, a total of 32,355 refined MAGs were retained, comprising 30,331 with medium quality and 2,024 with high quality.

[Figure 2]

Fig. 2 | Environmental and geographical distribution of metagenome-assembled genomes and their species-level clustering. **a**, A total of 32,355 MAGs recovered from 498 geographically and environmentally diverse microbial community samples in the TPMC. All MAGs have a completeness of at least 50% and a contamination level of less than 5%. **b**, Distribution of quality metrics for the MAGs, showing minimum, first quartile, median, third quartile, and maximum values. **c**, TPMC MAGs are clustered based on 95% ANI and 30% AF, resulting in 10,723 genomespecies. Representative genomes from GTDB R214 (n = 85,205), GEM (n = 22,746), TG2G (n = 3,241), and TARA (n = 957) were then included in the clustering to identify the novelty of the TPMC genomespecies. **d**, Geographical and environmental distribution of TPMC genomespecies. Region or water ecosystem labels are assigned to genomespecies that are clustered by MAGs from diverse regions or water ecosystems. The genomespecies that contained MAGs recovered solely from a specific ecosystem were annotated by letters. **e**, The majority of TPMC genomespecies with >1 MAG are restricted to individual water ecosystems and regions (Tibet or Qilian). MAG, metagenome-assembled genome; ANI, average nucleotide identity; AF, alignment fraction.

[Figure 3]

Fig. 3 | Novelty and phylogenomic distribution of the metagenome-assembled genomes and their biosynthetic potential. a, A phylogenetic tree built for all the TPMC genomespecies (n = 10,723) based on a concatenated alignment of 120 universally distributed bacterial single-copy genes and placement of each MAG in the GTDB-Tk reference tree. Two classes of the phylum Pseudomonadota are marked in the tree. The color of the branches indicates the phylum of the genomespecies. The outer layers indicate, for each genomespecies, the region and water ecosystem labels based on its MAGs, the count of MAGs, the largest count of BGCs among its MAGs, and the novelty (compared to the representative genomes of GTDB, GEM, and TG2G catalogs, respectively). **b** and **c**, BGCs (n = 73,864) are clustered into GCFs (n = 18,414) and GCCs (n = 2,681), with their novelty and the total number across different types presented. **d**, MAGs with more than 20 BGCs are labeled in (a), and the count of BGCs across different types of these MAGs is presented. MAG, metagenome-assembled genome; BGC, biosynthetic gene cluster; GCF, gene cluster family; GCC, gene cluster clan.

[Figure S4]

a

b

Fig. S4 | Environmental and geographical distribution of metagenome-assembled genomes and biosynthetic gene clusters. a, The relative frequency of BGC types across the phyla. The environmental and geographical distribution, and the proportion of the MAGs at the phylum level. Phyla with <1% number of MAGs were designated as “Others”. **b,** Environmental and geographical distribution, and the proportion of the BGC types. BGC, biosynthetic gene cluster; MAG, metagenome-assembled genome.

[Table S3]

Table S3 Characteristics of the metagenome-assembled genomes in the Tibetan Plateau aquatic genome and gene catalog.

Genome ID (ANI 0.95 of OTU representative sequences are marked with *)	Sample ID	Genomespecies (dRep OTU_ID, ANI 0.95)	Water ecosystem	Region	Num_contigs	N50	Assembly_Size	GC	Completeness	Contamination	Strain heterogeneity
2019_bqzb-1_bin.10.purify*	2019_bqzb-1	2376_0	River	Tibet	519	6150	2541731	0.33878	79.38	3.52	72
2019_bqzb-1_bin.13.purify*	2019_bqzb-1	6904_1	River	Tibet	107	49097	3165745	0.62115	84.21	1.89	14.29
2019_bqzb-1_bin.14.purify*	2019_bqzb-1	3546_0	River	Tibet	529	3345	1640940	0.43342	79.13	2.08	100
2019_bqzb-1_bin.2.purify*	2019_bqzb-1	352_0	River	Tibet	19	119586	1171239	0.38222	82.02	1.12	100
2019_bqzb-1_bin.23.purify	2019_bqzb-1	1715_1	River	Tibet	591	8531	3766984	0.37543	76.67	2.6	0
2019_bqzb-1_bin.24.purify*	2019_bqzb-1	7186_0	River	Tibet	980	3502	3113623	0.6364	60.14	3.51	50

Due to the large size of this table, only part of it is displayed here. Please see the full table in the Supplementary tables.xlsx.

[Table S4]

Table S4 Summary of the quality of MAGs grouped by ecosystems and regions.

		Saline lake	Freshwater lake	River	Hot spring	Wetland	Glacier
Number of contigs	Min	1	3	3	7	2	11
	Q1	141	149	191	209	190	236
	Median	237	258	354	390	381	446
	Average	314.4706	332.3713	426.4156	446.3977	465.6194	532.7919
	Q3	403	440	583.5	608.5	639	734.5
	Q4	3094	3464	2777	2340	4553	2457
N50	Min	1213	1416	1371	1228	1313	1541
	Q1	6720.5	6223	4253.5	3612.5	4371	3857
	Median	13846	12799	7505	6550	7781	7117.5
	Average	27227.86	27560.27	20216.89	17500.43	22906.68	22735.62
	Q3	30507.5	27821	15694	15148	17775	16677.25
	Q4	1608524	2801216	963661	571414	2642365	759327
	Min	312364	310889	308173	321333	379451	468083
	Q1	1580470	1629240	1550172	1622330	1690298	2101960

Due to the large size of this table, only part of it is displayed here. Please see the full table in the Supplementary tables.xlsx.

[Table S5]

Table S5 The comparison between the representative TPMC MAGs to the representative genomes from GTDB, GEM, TG2G, and TARA catalogs.

10,723 representative TPMC MAGs compared with 85,205 representative genomes of GTDB R214 (896 (8.4%) matched).						
TPMC representative MAG	Matched reference MAG	ANI	Coverage	TPMC OTU	Taxonomic annotation	TPMC ecosystem type
2020_DTH4-W1_bin.51.purify	GCA_003671255.1_genomic	0.967328	0.981404959	3796_3	d__Bacteria;p__Planctomycetota;c__Planctomycetia;o__Gemmatales;f__Gemmataceae;g__UBA969;s__UBA969 sp003671255	River
2019_bqzb-4_bin.52.purify	GCA_021738605.1_genomic	0.970294	0.977232925	2450_1	d__Bacteria;p__Bacteroidota;c__Bacteroidia;o__Chitinophagales;f__Chitinophagaceae;g__SXYR01;s__SXYR01 sp021738605	River
2020_CRC-20W-1-14m-1_bin.48.purify	GCA_945872875.1_genomic	0.987787	0.967346939	2301_1	d__Bacteria;p__Bacteroidota;c__Bacteroidia;o__Flavobacteriales;f__Flavobacteriaceae;g__Flavobacterium;s__Flavobacterium psychrophilum_N	Freshwater lake
2020_KMC-20W-1-20m-1_bin.204.purify	GCF_017355915.2_genomic	0.999833	0.965401249	3892_1	d__Bacteria;p__Bacteroidota;c__Bacteroidia;o__Cytophagales;f__Spirosomaceae;g__Tolluribacter;s__Tolluribacter sp017355915	Freshwater lake
H4-1_bin.20.purify	GCF_000828655.1_genomic	0.985647	0.964232489	4293_1	d__Bacteria;p__Thermotogota;c__Thermotoga;o__Thermotogales;f__DSM-5069;g__Pseudothermotoga_A;s__Pseudothermotoga_A caldifontis	Hot spring
2021-CBL-W3_bin.99.purify	GCA_009703465.1_genomic	0.981693	0.958715596	2851_2	d__Bacteria;p__Actinomycetota;c__Actinomycetia;o__Actinomycetales;f__Microbacteriaceae;g__Rhodoluna;s__Rhodoluna sp009703465	Wetland
2021-YM22-W1_bin.1.purify	GCA_018402665.1_genomic	0.962063	0.955974843	2017_1	d__Bacteria;p__Bacteroidota;c__Bacteroidia;o__Flavobacteriales;f__Schleiferiaceae;g__UBA10364;s__UBA10364 sp018402665	Saline lake
2020_KMC-20W-1-20m-1_bin.8.purify	GCA_903827085.1_genomic	0.97524	0.95505618	4884_1	d__Bacteria;p__Actinomycetota;c__Acidimicrobia;o__Acidimicrobiales;f__UBA8139;g__F1-20-MAGs160;s__F1-20-MAGs160 sp903827085	Freshwater lake

Due to the large size of this table, only part of it is displayed here. Please see the full table in the Supplementary tables.xlsx.

2. Within the large collection of more than 33k MAGs recovered in this study, 135 MAGs were classified as Archaea, according to Supplementary Table S3. Information about this intriguing group of organisms should be provided in the main text.

Answers: Many thanks for your comments. We have added the information about the Archaeal genomes to the main text (Lines 209 to 211). Owing to the refinement process above, the number of archaeal genomes has changed to 131.

[Main text]

Lines 209 to 211: Moreover, besides bacterial genomes (n = 32,224, 99.6%), we recovered 131 (0.4%) archaeal genomes, dominated by phyla Thermoproteota (n = 65), Halobacteriota (n = 24), and Nanoarchaeota (n = 16), and genus *Methanotherix* (n = 11) (Supplementary Table 3).

3. Viruses are known to play key roles in microbial population dynamics and nutrient cycling within aquatic ecosystems. The present study completely ignores these entities. I recommend including a section about viruses recovered from metagenomes sequenced in this study.

Answers: Many thanks for your comments. We appreciate your suggestion regarding the inclusion of a section on viruses in our study. However, the comprehensive establishment of virus genome resources requires large computing resources, dedicated software, and pipelines, which deserve a separate paper to. We have initiated the necessary processes on our server and plan to delve deeper into virus genomes in our subsequent work. In the revised manuscript, we have introduced virus detection of bin contigs, as outlined in Lines 580 to 582, aligning with your earlier comments (**Lines 580 to 582**). We appreciate your insightful suggestion and look forward to addressing viruses comprehensively in our future research.

[Main text]

Lines 580 to 582: Viral contigs were identified by VirSorter2 (version 2.2.4)¹², with parameters “--include-groups dsDNAphage, NCLDV, RNA, ssDNA, lavidaviridae, --min-length 5000, and--min-score 0.5”.

4. The authors use a very limited number of external datasets (GEM, TG2G, TARA oceans) to compare and define the novelty of their recovered MAGs. Including more datasets, especially from similar environments, would strengthen the validity of findings presented in this study.

Answer: Many thanks for your comments. In response to your suggestion, we have expanded our dataset by incorporating 85,205 representative genomes from the GTDB R214 and found 894 (8.4%) TPMC genomespecies mapped to them. This addition aims to provide a more comprehensive comparison and a robust assessment of the novelty of the recovered MAGs in our study. We have fully updated these results in the main text (**Lines 169 to 172**), **Figure 2c**, **Figure 3a**, **Figure S1**, and **Table S5**.

[Main text]

Lines 169 to 172: The 10,723 genomespecies demonstrated substantial novelty, with 91.64%, 98.46%, 98.78%, and 99.98% exhibiting low sequence identity to 85,205 GTDB genomes, 22,746 GEM genomes, 3,241 TG2G genomes, and 957 TARA genomes, respectively (**Supplementary Table 5**).

[Figure 2c]

Fig. 2 | Environmental and geographical distribution of metagenome-assembled genomes and their species-level clustering. c, TPMC MAGs are clustered based on 95% ANI and 30% AF, resulting in 10,723 genomespecies. Representative genomes from GTDB R214 (n = 85,205), GEM (n = 22,746), TG2G (n = 3,241), and TARA (n = 957) were then included in the clustering to identify the novelty of the TPMC genomespecies.

[Figure 3a]

Only part of Figure 3a is displayed here for review.

Fig. 3 | Novelty and phylogenomic distribution of the metagenome-assembled genomes and their biosynthetic potential. a, A phylogenetic tree built for all the TPMC genomespecies (n = 10,723) based on a concatenated alignment of 120 universally distributed bacterial single-copy genes and placement of each MAG in the GTDB-Tk reference tree. Two classes of the phylum Pseudomonadota are marked in the tree. The color of the branches indicates the phylum of the genomespecies. The outer layers indicate, for each genomespecies, the region and water ecosystem labels based on its MAGs, the count of MAGs, the largest count of BGCs among its MAGs, and the novelty (compared to the representative genomes of GTDB, GEM, and TG2G catalogs, respectively).

[Figure S1]

Only part of Figure S1 is displayed here for review.

Fig. S1 | Overview of the metagenomic analysis.

[Table S5]

Table S5 The comparison between the representative TPMC MAGs to the representative genomes from GTDB, GEM, TG2G, and TARA catalogs.

10,723 representative TPMC MAGs compared with 85,205 representative genomes of GTDB R214 (896 (8.4%) matched).						
TPMC representative MAG	Matched reference MAG	ANI	Coverage	TPMC OTU	Taxonomic annotation	TPMC ecosystem type
2020_DTH4-W1_bin.51_purify	GCA_003671255.1_genomic	0.967328	0.981404959	3786_3	d__Bacteria;p__Planctomycetota;c__Planctomycetia;o__Gemmatales;f__Gemmataceae;g__UBA969;s__UBA969 sp003671255	River
2019_bqzb-4_bin.52_purify	GCA_021738605.1_genomic	0.970294	0.977232925	2450_1	d__Bacteria;p__Bacteroidota;c__Bacteroidia;o__Chitinophagales;f__Chitinophagaceae;g__SXYR01;s__SXYR01 sp021738605	River
2020_CRC-20W-1-14m-1_bin.48_purify	GCA_945872875.1_genomic	0.987787	0.967346939	2301_1	d__Bacteria;p__Bacteroidota;c__Bacteroidia;o__Flavobacteriales;f__Flavobacteriaceae;g__Flavobacterium;s__Flavobacterium psychrophilum_N	Freshwater lake
2020_KMC-20W-1-20m-1_bin.204_purify	GCF_017355915.2_genomic	0.999833	0.965401249	3892_1	d__Bacteria;p__Bacteroidota;c__Bacteroidia;o__Cytophagales;f__Spirosomaceae;g__Tetraubacter;s__Tetraubacter sp017355915	Freshwater lake
H4-1_bin.20_purify	GCF_000828655.1_genomic	0.985647	0.964232489	4293_1	d__Bacteria;p__Thermotogota;c__Thermotogales;o__Thermotogales;f__DSM-5069;g__Pseudothermotoga_A;s__Pseudothermotoga_A caldfontis	Hot spring
2021-CBL-W3_bin.99_purify	GCA_009703465.1_genomic	0.961693	0.958715596	2851_2	d__Bacteria;p__Actinomycetota;c__Actinomycetia;o__Actinomycetales;f__Microbacteriaceae;g__Rhodoluna;s__Rhodoluna sp009703465	Wetland
2021-YMQ2-W1_bin.1_purify	GCA_018402665.1_genomic	0.962063	0.955974843	2017_1	d__Bacteria;p__Bacteroidota;c__Bacteroidia;o__Flavobacteriales;f__Schlieffeniaceae;g__UBA10364;s__UBA10364 sp018402665	Saline lake
2020_KMC-20W-1-20m-1_bin.8_purify	GCA_903827085.1_genomic	0.97524	0.95505618	4884_1	d__Bacteria;p__Actinomycetota;c__Acidimicrobia;o__Acidimicrobiales;f__UBA8139;g__F1-20-MAGs160;s__F1-20-MAGs160 sp903827085	Freshwater lake

Due to the large size of this table, only part of it is displayed here. Please see the full table in Supplementary tables.xlsx.

5. Some previous metagenomic studies from the Tibetan Plateau were not acknowledged (for example: <https://doi.org/10.1016/j.envres.2022.114847>, <https://doi.org/10.1016/j.isci.2021.103439>).

Overall, the authors make very few references to past studies from this same geographical region and do not properly integrate their own findings and observations with past work.

Answer: Many thanks for your comments. In response, we have duly acknowledged more previous metagenomic studies in our revised manuscript (Lines 66 to 77 and Lines 202 to 205). It's important to note that our focus on catalog establishment, especially the comprehensive establishment of MAGs, led us to mainly compare our results with previous work that involved the establishment of the TG2G in TP⁴.

[Main text]

Lines 66 to 77: For metagenome-assembled genomes (MAGs) in TP metagenomes, Liu et al. successfully recovered 2,358 MAGs from 85 snow, ice, and cryoconite metagenomes of TP glaciers⁴. Wei et al. also contributed to this field by recovering 75 MAGs from two soil and water metagenomes of a TP saline lake²⁰, while Yun et al. obtained 200 MAGs from two soil samples of Tibet wetlands²¹. Additionally, Hu et al. recovered 278 MAGs from 69 water and sediment metagenomes of TP wetlands and rivers²². Notably, among these studies, only the TP glacier catalog has been systematically established by Liu et al⁴; the other studies did not have the specific goal of establishing a resource. It is crucial to emphasize the imperative need to catalog microbial resources in various ecosystems, including rivers, due to their essential role in the hydrological cycle and their connections to human societies. Wetlands are also of paramount importance, given that approximately 80% of global wetland resources are degrading or disappearing²³.

Lines 202 to 205: The dominance of Pseudomonadota (synonym Proteobacteria), Bacteroidota, and Actinomycetota was consistent with previous studies in TP saline lakes and rivers^{20,22}. However, we recovered more MAGs from Verrucomicrobiota, and it took the place of Bacillota (synonym Firmicutes), which dominated previous studies^{20,22}.

6. The abundance of the non-redundant gene catalog was calculated, however, there is no information provided regarding the most abundant organisms at the species level and no characterization of these organisms.

Answers: Many thanks for your comments. In response, we have provided the most abundant organisms at the phylum and species levels in the revised manuscript (**Lines 100 to 106 and Figure S2ab**).

[Main text]

Lines 100 to 106: The analysis revealed that the microbial community composition was dominated by the phyla Pseudomonadota, Bacteroidota, and Actinomycetota (**Supplementary Fig. 2a**). Notably, *Cyanobium usitatum* was among the most abundant species in saline lakes, freshwater lakes, rivers in Tibet, and saline lakes in Qilian (**Supplementary Fig. 2b**). *Acinetobacter johnsonii* featured prominently in rivers in Qilian, while *Polynucleobacter duraquae* dominated wetlands in both Tibet and Qilian. Additionally, *Tepidimonas fonticaldi* was the most abundant species in hot springs in Tibet.

[Figure S2ab]

Fig. S2 | Microbial composition, assemblage, and networks of the Tibetan Plateau microbial communities. **a**, The relative abundances of the top 10 phyla across water ecosystems and regions. **b**, Boxplots show the relative abundances of the top 3 species across water ecosystems and regions.

7. L440 “The taxonomic abundance profile of FMLG microbial community” - the FMLG community is not defined anywhere in the text.

Answer: Many thanks for your comments. This is a typo, and it should have been TPMC. We have modified the sentences for clarity.

[Main text]

Lines 487 to 488: The taxonomic abundance profile of the TPMC microbial community for diversity analysis was obtained by MetaPhlan (version 4.0.1)¹⁰ with default parameters based on metagenome reads.

Minor/Miscellaneous points:

8. The phylogenetic tree produced by GTDB-Tk with the classify_wf (Figure 3a) is not generated de novo but by inserting concatenated markers recovered from MAGs in this study in an existing phylogenetic tree with pplacer. The authors should make this clear in the methods and figure legend.

Answer: Many thanks for your comments. We have added this information to the methods and figure legend for clarity (**Lines 605 to 607 and Figure 3a**).

[Main text]

Lines 605 to 607: The phylogenetic tree was generated through maximum-likelihood placement of each MAG of this study in the GTDB-Tk reference tree using pplacer²⁴.

Lines 926 to 928 | Figure 3a: A phylogenetic tree built for all the TPMC genomespecies (n=10,723) based on a concatenated alignment of 120 universally distributed bacterial single-copy genes and placement of each MAG in the GTDB-Tk reference tree.

8. It would be preferable that individual classes be highlighted within Proteobacteria on the phylogenetic tree since this is a vast phylum.

Answer: Many thanks for your comments. In response, we have highlighted the classes within phylum Pseudomonadota (synonym Proteobacteria) in **Figure 3a**. All phylum names have been revised to align with the taxonomic annotations of the GTDB r214, ensuring uniformity and accuracy.

[Figure 3a]

Only part of the figure is displayed here for review.

3. The use of English throughout the manuscript, especially the Material and Methods section, as well as the title itself, requires improvements. Having a native English speaker check the manuscript could help improve the quality.

Answer: Many thanks for your comments. To address this concern, we have refined the language throughout the manuscript, focusing on clarity and grammatical accuracy. While numerous changes have been made, we have highlighted the most significant ones. Furthermore, we have changed the title to “A genome and gene catalog of Tibetan Plateau's aquatic microbiomes”.

Reviewer #3

The paper describes over 11,000 novel microbial species, ~60 million novel gene clusters and 35,000 novel biosynthetic gene clusters. 498 metagenomes from diverse Tibetan Plateau water ecosystems enabled comparative analysis of beta diversity, revealing negative correlations of similarity with distance and altitude, supporting the idea of a “microbial watertower”, dispersing microbes throughout the region.

In my opinion, the work is significant to the field, with well-supported conclusions. While I did not find any major flaws, I request re-analysis with updated GTDB (r214 released April 2023) and better taxonomic consistency (see below). Minor comments below.

Answer: Many thanks for your comments. In response to your suggestion, we have re-analyzed the data using the updated GTDB (R214), and we have ensured that all taxonomic annotations are consistent with GTDB (R214). For instance, Proteobacteria has been changed to Pseudomonadota used in GTDB R214. Additionally, we have updated all related results, figures, and tables accordingly, and listed several changes here for review:

[Main text]

Lines 194 to 202: Utilizing taxonomic annotations from the Genome Taxonomy Database (GTDB release R214)²⁵, we identified a wide spectrum of taxonomic diversity within the MAGs, covering 83 known phyla, 186 known classes, 470 known orders, 952 known families, 1,835 known genera, and 993 known species (**Supplementary Table 3**). Subsequently, we constructed a phylogeny of the 10,723 genomespecies based on 120 concatenated bacterial marker genes, highlighting the top 11 phyla containing more than 1% of the total MAGs (**Fig. 3a** and **Supplementary Fig. 4a**). Among these phyla, the MAG catalog was predominantly composed of Pseudomonadota (n=10,183, 31.5%), Bacteroidota (n=7,551, 23.3%), Actinomycetota (n=6,163, 19.0%), and Verrucomicrobiota (n=1,993, 6.2%).

Lines 603 to 605: The taxonomy annotation of the 10,723 genomespecies was performed using the module “classify_wf” of Genome Taxonomy Database Toolkit (GTDB-Tk, version 2.2.6)¹⁸ against the GTDB release R214²⁵ with default parameters.

Table S3 Characteristics of the metagenome-assembled genomes in the Tibetan Plateau aquatic genome and gene catalog.

GTDB_classification
d__Bacteria;p__Bacteroidota;c__Bacteroidia;o__Flavobacteriales;f__Flavobacteriaceae;g__Flavobacterium;s__
d__Bacteria;p__Pseudomonadota;c__Gammaproteobacteria;o__Burkholderiales;f__Burkholderiaceae_B;g__Hydrogenophaga;s__
d__Bacteria;p__Pseudomonadota;c__Gammaproteobacteria;o__Burkholderiales;f__Methylophilaceae;g__Methylotenera_A;s__
d__Bacteria;p__Patescibacteria;c__JAEDAM01;o__BD1-5;f__UBA2023;g__JABFSX01;s__

Due to the large size of this table, only part of it is displayed here. Please see the full table in Supplementary tables.xlsx.

[Figure 2]

Fig. 2 | Environmental and geographical distribution of metagenome-assembled genomes and their species-level clustering. **a**, A total of 32,355 MAGs recovered from 498 geographically and environmentally diverse microbial community samples in the TPMC. All MAGs have a completeness of at least 50% and a contamination level of less than 5%. **b**, Distribution of quality metrics for the MAGs, showing minimum, first quartile, median, third quartile, and maximum values. **c**, TPMC MAGs are clustered based on 95% ANI and 30% AF, resulting in 10,723 genomespecies. Representative genomes from GTDB R214 (n = 85,205), GEM (n = 22,746), TG2G (n = 3,241), and TARA (n = 957) were then included in the clustering to identify the novelty of the TPMC genomespecies. **d**, Geographical and environmental distribution of TPMC genomespecies. Region or water ecosystem labels are assigned to genomespecies that are clustered by MAGs from diverse regions or water ecosystems. The genomespecies that contained MAGs recovered solely from a specific ecosystem were annotated by letters. **e**, The majority of TPMC genomespecies with >1 MAG are restricted to individual water ecosystems and regions (Tibet or Qilian). MAG, metagenome-assembled genome; ANI, average nucleotide identity; AF, alignment fraction.

[Figure 3a]

Fig. 3 | Novelty and phylogenomic distribution of the metagenome-assembled genomes and their biosynthetic potential. a, A phylogenetic tree built for all the TPMC genomespecies (n = 10,723) based on a concatenated alignment of 120 universally distributed bacterial single-copy genes and placement of each MAG in the GTDB-Tk reference tree. Two classes of the phylum Pseudomonadota are marked in the tree. The color of the branches indicates the phylum of the genomespecies. The outer layers indicate, for each genomespecies, the region and water ecosystem labels based on its MAGs, the count of MAGs, the largest count of BGCs among its MAGs, and the novelty (compared to the representative genomes of GTDB, GEM, and TG2G catalogs, respectively).

1. Line 81: Remove ever

Answer: Many thanks for your comments. We have removed “ever” and modified the sentences (**Lines 86 to 88**).

Lines 86 to 88: The TPMC catalog stands out as the largest and most comprehensive repository of aquatic microbial resources on the TP, covering diverse water ecosystems.

2. Line 127: “The” instead of “he”

Answer: Many thanks for your comments. We have revised the typo and modified the sentences (**Lines 145 to 149**).

Lines 145 to 149: We performed metagenomic assembly and binning on the 498 metagenomes and recovered a total of 32,355 MAGs. These MAGs met or exceeded the medium-quality criteria outlined in the Minimum Information about a Metagenome-Assembled Genome (MIMAG) standard¹⁶, with mean completeness of 78.1% and mean contamination of 2.6%.

3. Lines 129/143: How many novel species had a MAG that passed quality score > 50 and also HQ?

Answer: Many thanks for your comments. A total of 9,698 genomespecies did not map to any of the genomes from GTDB R214, GEM, TG2G, and TARA databases, of which 6,335 passed a quality score >50, and 465 met high-quality criteria (Also passed a quality score >50). We have added this information to the revised manuscript (**Lines 169 to 175**).

[Main text]

Lines 169 to 175: The 10,723 genomespecies demonstrated substantial novelty, with 91.64%, 98.46%, 98.78%, and 99.98% exhibiting low sequence identity to 85,205 GTDB genomes, 22,746 GEM genomes, 3,241 TG2G genomes, and 957 TARA genomes, respectively (**Supplementary Table 5**). A total of 9,698 genomespecies did not map to any of these catalog genomes, of which 6,335 passed a quality score >50, and 465 met high-quality criteria. In particular, 219 out of 245 genomespecies from the TP glacier did not map to the TG2G genomes.

4. Lines 146: How many OTUs were novel to both? How many are novel when compared to GTDB?

Answer: Many thanks for your comments. A total of 9,698 genomespecies did not map to any of the genomes from GTDB R214, GEM, TG2G, and TARA databases (**Lines 169 to 175**).

[Main text]

Lines 169 to 175: The 10,723 genomespecies demonstrated substantial novelty, with 91.64%, 98.46%, 98.78%, and 99.98% exhibiting low sequence identity to 85,205 GTDB genomes, 22,746 GEM genomes, 3,241 TG2G genomes, and 957 TARA genomes, respectively (**Supplementary Table 5**). A total of 9,698 genomespecies did not map to any of these catalog genomes, of which 6,335 passed a quality score >50,

and 465 met high-quality criteria. In particular, 219 out of 245 genomespecies from the TP glacier did not map to the TG2G genomes.

5. Line 147: “In particular” instead of “Specifically”

Answer: Many thanks for your comments. We have replaced the word “Specifically” with the word “In particular” (Lines 174 to 175).

[Main text]

Lines 174 to 175: In particular, 219 out of 245 genomespecies from TP glacier did not map to the TG2G genomes.

6. Line 152: How many mapped?

Answer: Many thanks for your comments. We have found two genomespecies mapping to the TARA database. We have added this information to the revised manuscript (Lines 169 to 175, Figure 2c, and Table S5).

[Main text]

Lines 169 to 175: The 10,723 genomespecies demonstrated substantial novelty, with 91.64%, 98.46%, 98.78%, and 99.98% exhibiting low sequence identity to 85,205 GTDB genomes, 22,746 GEM genomes, 3,241 TG2G genomes, and 957 TARA genomes, respectively (Supplementary Table 5). A total of 9,698 genomespecies did not map to any of these catalog genomes, of which 6,335 passed a quality score >50, and 465 met high-quality criteria. In particular, 219 out of 245 genomespecies from the TP glacier did not map to the TG2G genomes.

[Figure 2c]

Fig. 2 | Environmental and geographical distribution of metagenome-assembled genomes and their species-level clustering. c, TPMC MAGs are clustered based on 95% ANI and 30% AF, resulting in 10,723 genomespecies. Representative genomes from GTDB R214 (n = 85,205), GEM (n = 22,746), TG2G (n = 3,241), and TARA (n = 957) were then included in the clustering to identify the novelty of the TPMC genomespecies.

[Table S5]

Table S5 The comparison between the representative TPMC MAGs to the representative genomes from GTDB, GEM, TG2G, and TARA catalogs. The results with ANI ≥ 0.95 were showed. MAGs with ANI ≥ 0.95 and Coverage ≥ 0.3 were designated as the matched MAGs.

10,723 representative TPMC MAGs compared with 957 representative MAGs of TARA ocean (2 (0.02%) matched).						
TPMC representative MAG	Matched reference MAG	ANI	Coverage	TPMC OTU	Taxonomic annotation	TPMC ecosystem type
2020_QHH7-W1_bin.125.purify	TARA_ION_MAG_00004	0.977303	0.905882353	6680_2	d__Bacteria.p__Pseudomonadota.c__Gammaproteobacteria.o__Burkholderiales.f__Burkholderiaceae.g__Limnobacter.s__Limnobacter sp.002954425	Saline lake
2020_TMRSD3-W1_bin.70.purify	TARA_ION_MAG_00002	0.971879	0.77027027	5929_1	d__Bacteria.p__Pseudomonadota.c__Alphaproteobacteria.o__Caulobacterales.f__Caulobacteraceae.g__Brevundimonas.s__Brevundimonas aurantica	Wetland

Due to the large size of this table, only part of it is displayed here. Please see the full table in Supplementary tables.xlsx.

7. Line 153: This conclusion could be improved by quantifying the contribution of TPMC. For instance, by reporting the increase in mapped % for all TP metagenomes when TPMC species representatives are added to the previous databases.

Answer: Many thanks for your comments. In response to your comment, we have conducted additional analysis by recruiting 85,205 representative genomes from GTDB R214 for comparison. The results indicate that only 896 (8.4%) TPMC genomes/species mapped to them. The novel TPMC genomes/species ($n = 9,827$) represent 11.5% of the 85,205 representative genomes from GTDB. This allows us to accurately state that we are "expanding the previous GTDB database by 11.5%." We have added this conclusion to the revised manuscript (**Lines 179 to 182 and Lines 385 to 387**).

[Main text]

Lines 179 to 182: It expands the previous GTDB database by 11.5% (9,827 novel genomes compared to 85,205 references) and the GEM database by 46.4% (10,558 novel genomes compared to 22,746 references), thus largely contributing to the completion of the global microbiome reservoir.

Lines 388 to 389: The TPMC significantly expands the GTDB genomic catalog by 11.5% and the NCBI non-redundant gene set by approximately fifty million in number.

8. Line 178: The new GTDB update (r214) includes 15 new phyla, etc. Can you repeat the analysis using the newest GTDBtk version?

Answer: Many thanks for your comments. In response to your suggestion, we have re-conducted the analysis using the latest GTDB R214 (**Lines 194 to 197**) and updated all the figures and tables accordingly (**Instances of changes have been listed under your first comment**). Noteworthy improvements resulting from the utilization of the updated GTDB database include a 34% reduction in the proportion of unclassified families, a 20% decrease in unclassified genera, and an 8.7% decrease in unclassified species. Although the changes in the phylum category were not statistically significant (still three unclassified phyla), this update has significantly enhanced the precision and applicability of the MAG resources for fellow researchers.

[Main text]

Lines 194 to 197: Utilizing taxonomic annotations from the Genome Taxonomy Database (GTDB release R214)²⁵, we identified a wide spectrum of taxonomic diversity within the MAGs, covering 83 known phyla, 186 known classes, 470 known orders, 952 known families, 1,835 known genera, and 993 known species (**Supplementary Table 3**).

9. Line 202: How many unigenes were not annotated by any of the databases?

Answer: Many thanks for your comments. A total of 46,670,736 (15.8%) unigenes were not annotated by any of the databases. In response, we have added this result to the main text (**Lines 243 to 244**).

[Main text]

Lines 243 to 244: A total of 46,670,736 (15.8%) unigenes remained unannotated by any of the databases, thus designated as novel genes.

10. Line 223: Please add a comment about why you think VF0082 is useful for pathogen monitoring, considering the prevalence of motility in non-pathogenic microbes.

Answer: Many thanks for your comments. We listed these VFs such as VF0082, VF0465, or VF0091 because they were the most prevalent VFs in each biome. We believe that the provided VF resource could be instrumental for VF monitoring. To avoid misunderstanding, we have rephrased the comments.

[Main text]

Lines 269 to 272: These findings highlight the varied distribution patterns of VFs across diverse water ecosystems in the TP. This knowledge could serve as a valuable resource for VF monitoring, aiding in the mitigation of potential risks to human health.

11. Line 224: “is” instead of “was”

Answer: Many thanks for your comments. We have replaced the word “was” with the word “is” in the sentences related to the VF description (**Lines 262 to 264**).

Lines 262 to 264: VF0082 is involved in transcriptional regulation and chemosensory pathways controlling the twitching motility of the pili, while VF0840 modulates small multidrug resistance (SMR) antibiotic efflux pump.

12. Line 274: Please also provide Pielou’s evenness, as a better alpha diversity metric for between group comparisons.

Answer: Many thanks for your comments. In response, we have updated the Figure S7 using the Pielou’s evenness.

[Main text]

Lines 311 to 313: Among all water ecosystems and regions, rivers and wetlands exhibited the highest

Pielou's evenness diversity of BGCs, GCFs, and GCCs, while hot springs from Tibet displayed the lowest diversity ($P < 0.01$, **Supplementary Fig. 7**).

[Figure S7]

Fig. S7 | Metagenomic profiles of the biosynthetic gene clusters. The Pielou's evenness diversity of BGCs, GCFs, and GCCs, with minimum, first quartile, median, third quartile and maximum values presented across the water ecosystems. BGC, biosynthetic gene cluster; GCF, gene cluster family; GCC, gene cluster clan.

13. Line 278: Comment on why Tibet/Qilian saline lake samples cluster separately.

Answers: Many thanks for your comments. The distinct biosynthetic potential observed in the Qilian saline lake samples may be attributed to 1) their unique microbial compositions, separated apart from other groups; and 2) their microbial assemblage exhibited a better fit to neutral theory than others. However, it still needs further investigation. We have incorporated the following comments into the results section (**Lines 317 to 322**).

[Main text]

Lines 317 to 322: The unique BGC compositions observed in Qilian Saline lakes may be linked to their distinct microbial compositions compared to other ecosystems (**Fig. 1b**). Furthermore, their microbial assemblage demonstrated a better fit to the neutral theory, suggesting a lesser degree of environmental

intervention in microbiome assemblage (**Supplementary Fig. 2c**). Further investigations are warranted to elucidate the mechanisms behind the formation of these unique BGC compositions in Qilian Saline lakes.

14. Line 287: Refers to Figure 3d instead of Figure 2d

Answer: Many thanks for your comments. We have fixed this typo (**Lines 327 to 329**).

[Main text]

Lines 327 to 329: Furthermore, we identified 11 BGC-rich genomespecies that contained at least one MAG with >20 BGCs (**Fig. 3d and Supplementary Table 13**), and their BGCs were enriched in NRPs and RiPPs clusters.

15. Line 287: Remove “found mostly”

Answer: Many thanks for your comments. We have removed the words (**Lines 327 to 329**).

[Main text]

Lines 327 to 329: Furthermore, we identified 11 BGC-rich genomespecies that contained at least one MAG with >20 BGCs (**Fig. 3d and Supplementary Table 13**), and their BGCs were enriched in NRPs and RiPPs clusters.

16. Line 288: The “c” MAGs mentioned here appear to be located in an “other” phyla in Figure 3a, not in Proteobacteria? Are their bin names mentioned somewhere, so that I can find them in Table S3? This may be a NCBI vs GTDB taxonomy issue, if so, then please use the same taxonomy throughout (e.g. switch completely to GTDB by referring to these “other” MAGs as Myxococcota or revert to NCBI by changing Bdellovibrionota to Proteobacteria, etc.).

Answer: Many thanks for your comments. In response to your comments, we have made the following revisions:

1) Addition of **Table S13**:

We have introduced **Table S13**, which provides information on the OTUs (referred to as genomespecies) containing a MAG with more than 20 predicted BGCs (**Lines 327 to 329**). This table includes the count of each type of BGCs, along with the MAG taxonomy. Moreover, in the updated **Figure 3a**, the “f” (“c” in the previous manuscript) MAGs are located in the phylum Myxococcota (Gray branches in the tree).

2) Consistent Use of GTDB R214 Taxonomy:

We have opted for uniformity in taxonomy throughout the manuscript, completely transitioning to GTDB R214. Specifically, Proteobacteria has been switched to the synonym Pseudomonadota in line with the latest GTDB R214.

[Main text]

Lines 327 to 329: Furthermore, we identified 11 BGC-rich genomespecies that contained at least one MAG with >20 BGCs (**Fig. 3d and Supplementary Table 13**) and found their BGCs were enriched in NRPs and RiPPs clusters.

[Figure 3a]

Only part of the figure is displayed here for review.

Fig. 3 | Novelty and phylogenomic distribution of the metagenome-assembled genomes and their biosynthetic potential. a, A phylogenetic tree built for all the TPMC genomespecies ($n = 10,723$) based on a concatenated alignment of 120 universally distributed bacterial single-copy genes and placement of each MAG in the GTDB-Tk reference tree. Two classes of the phylum Pseudomonadota are marked in the tree. The color of the branches indicates the phylum of the genomespecies. The outer layers indicate, for each genomespecies, the region and water ecosystem labels based on its MAGs, the count of MAGs, the largest count of BGCs among its MAGs, and the novelty (compared to the representative genomes of GTDB, GEM, and TG2G catalogs, respectively).

[Table S13]

Table S13 The genomespecies presenting the highest biosynthetic potential. The genomespecies that contained a MAG with more than 20 predicted BGCs are listed here.

MAG	Genomespecies (dRep OTU_ID, ANI 0.95)	BGC Sum	Terpene	RiPPs	Others	PKSother	NRPS	PKSI	PKS-NRP_Hybrids	Saccharides	taxonomy
2020_CBL-wt_bin.10.purify	5408_0	84	3	7	6	4	47	1	16	0	d__Bacteria_p__Myxococcota_c__Myxococcia_o__Myxococcales_f__Myxococaceae_g__Corallococcus_s__Corallococcus_sicarius
2021-BC-W20_bin.87.purify	5407_0	80	2	11	2	2	42	7	14	0	d__Bacteria_p__Myxococcota_c__Myxococcia_o__Myxococcales_f__Myxococaceae_g__Myxococcus_s__Myxococcus_caerfyddinensis
2021-OTH-W5_bin.24.purify	3871_1	48	7	7	6	2	9	12	5	0	d__Bacteria_p__Actinomycetota_c__Actinomycetia_o__Streptomycetales_f__Streptomycetaceae_g__Streptomyces_s__Streptomyces_ahisflavus
2020_CBL-w3_bin.99.purify	3750_1	32	3	6	0	1	12	5	5	0	d__Bacteria_p__Chloroflexota_c__Anaerolineae_o__Caldilineales_f__Caldilineaceae_g__JAHBVH01_s__
2020_YBJS-DW-3_bin.35.purify	2793_0	a	0	8	2	2	11	0	3	0	d__Bacteria_p__Cyanobacteriota_c__Cyanobacteria_o__Cyanobacteriales_f__Nostocaceae_g__Tichoniusus_s__Tichoniusus_sp022063185
2021-BBH-W2_bin.31.purify	7530_1	25	0	5	5	2	11	1	1	0	d__Bacteria_p__Pseudomonadota_c__Gammaproteobacteria_o__Pseudomonadales_f__Pseudomonadaceae_g__Pseudomonas_E_s__Pseudomonas_E_protegens_A
2020_CBL-w2_bin.167.purify	1140_0	24	2	7	3	0	5	5	2	0	d__Bacteria_p__Myxococcota_c__Polyanglia_o__Polyangliales_f__Polyangliaceae_g__Polyangium_s__
2020_NMC-W8-15m-1_bin.163.purify	5201_0	24	3	1	3	0	16	1	0	0	d__Bacteria_p__Actinomycetota_c__Actinomycetia_o__Mycobacteriales_f__Mycobacteriaceae_g__Rhodococcus_s__Rhodococcus_pyridinivorans
2020_SYH2-W1_bin.24.purify	5032_1	22	2	2	2	1	9	4	2	0	d__Bacteria_p__Actinomycetota_c__Actinomycetia_o__Mycobacteriales_f__Mycobacteriaceae_g__Mycobacterium_s__Mycobacterium_bacteremicum
2019_TBMTW-1_bin.25.purify	7349_0	21	1	3	3	3	10	0	1	0	d__Bacteria_p__Pseudomonadota_c__Gammaproteobacteria_o__Xanthomonadales_f__Xanthomonadaceae_g__SCMT01_s__
2021-HSDK-W2_bin.24.purify	7646_2	21	3	7	0	0	10	1	0	0	d__Bacteria_p__Cyanobacteriota_c__Cyanobacteria_o__Cyanobacteriales_f__Oscillatoriales_g__Oscillatoria_s__

17. Line 301: “adaptation” instead of “adaption”

Answer: Many thanks for your comments. We have fixed this typo.

Lines 342 to 343: These BGC-rich species could be strong candidates for exploring the underlying mechanisms of microbes in adaptation to the extreme environment of the TP.

18. Line 325: “representing one of” instead of “representing the one of”

Answer: Many thanks for your comments. We have replaced the words “representing the one of” with the words “representing one of” (**Lines 366 to 367**).

Lines 366 to 367: ...representing one of the largest gene catalogs in a single microbiome study.

19. Line 332: “biogeography” typo

Answer: Many thanks for your comments. We have fixed this typo (**Lines 373 to 374**).

Lines 373 to 374: ...environmental factors also influenced TP microbiome biogeography...

20. Line 342: Sentence is a bit clunky, please reword.

Answer: We have reworded the sentences for clarity (**Lines 380 to 383**).

Lines 380 to 383: These findings revealed a potential dispersal pattern in microbiome biogeography. For various water ecosystems situated at greater geographical distances and altitude differences from the TP, their microbiomes may exhibit fewer compositional and functional similarities with the TP microbiome, underscoring the potential role of the TP as a "microbial tower."

21. Line 357: “and store”: Is this referring to frozen sample storage? It suggests cultured organisms.

Answer: Many thanks for your comments. The word "store" refers to cataloging microbial resources

rather than frozen sample storage. To eliminate any ambiguity, we have replaced the term with “catalog” (Lines 397 to 400).

[Main text]

Lines 397 to 400: Furthermore, in the context of TP suffering substantial water loss due to global warming, TPMC is the first attempt to resolve and catalog the aquatic microbial resources across a range of water ecosystems in TP, facilitating the preservation of the TP water resource.

22. Line 376: “Everything is everywhere” does not suggest that we have isolated genomes for everything. I don’t think there is a contradiction here.

Answer: Many thanks for your comments. We have removed the related discussion to eliminate any potential confusion or misunderstanding (Lines 416 to 419).

[Main text]

Lines 416 to 419: Furthermore, for biogeography on a global scale^{26,27}, our TP genome catalog exhibited only a small proportion of shared genomes (1.5%) with the GEM genome catalog²⁸, underscoring TP’s unique microbial resources and their distinct responses to extreme environmental stresses.

23. Line 386: “advancing” sounds too direct. You are providing data that may contribute to these areas, not directly advancing them.

Answer: Many thanks for your comments. We have replaced the word “advancing” with the word “contributes to” (Lines 426 to 428).

Lines 426 to 428: This study not only bridges knowledge gaps in the global microbial reservoir but also contributes to biotechnological applications in enzymology, environmental protection, and healthcare.

24. Methods: please specify how the “similarity” was calculated for Figure 5.

Answer: Many thanks for your comments. We have specified how “similarity” was calculated in the Figure 5.

[Main text]

Lines 958 to 962: In all scatter plots, each dot represents a pairwise comparison in 1) microbial similarity quantified by the Bray–Curtis distance of species profiles produced by MetaPhlAn (version 4.0.1)¹⁰; 2) shared gene count; 3) geographical distance; and 4) altitude, which were conducted on samples within the TP (first ladder step), as well as between the TP and the other two ladder steps.

25. Table S3: please report GTDBtk notes and warnings columns

Answers: Many thanks for your comments. We have incorporated the "GTDB note" and "warnings"

columns into **Table S3** to provide comprehensive information.

[**Table S3**]

Table S3 Characteristics of the metagenome-assembled genomes in the Tibetan Plateau aquatic genome and gene catalog.

GTDB_Note	GTDB_Warnings
classification based on placement in class-level tree	Genome not assigned to closest species as it falls outside its pre-defined ANI radius
classification based on placement in class-level tree	Genome not assigned to closest species as it falls outside its pre-defined ANI radius
classification based on placement in class-level tree	Genome not assigned to closest species as it falls outside its pre-defined ANI radius
classification based on placement in class-level tree	N/A
classification based on placement in class-level tree	Genome not assigned to closest species as it falls outside its pre-defined ANI radius
classification based on placement in class-level tree	Genome not assigned to closest species as it falls outside its pre-defined ANI radius
classification based on placement in class-level tree	Genome not assigned to closest species as it falls outside its pre-defined ANI radius
classification based on placement in class-level tree	Genome not assigned to closest species as it falls outside its pre-defined ANI radius

Due to the large size of this table, only part of it is displayed here. Please see the full table in the Supplementary tables.xlsx.

26. Figure S7: Why is the Qilian glacier boxplot orange in the first plot, but green in the others?

Answer: Many thanks for your comments. We acknowledge that this was an oversight in the figure preparation. We have rectified the mistake and ensured uniform color representation throughout **Figure S7**. The Qilian glacier boxplot is now consistently presented with the appropriate color.

[**Figure S7**]

Fig. S7 | Metagenomic profiles of the biosynthetic gene clusters.

References

- 1 Qiu, J. China: The third pole. *Nature* **454**, 393–396 (2008). <https://doi.org/10.1038/454393a>
- 2 Yao, T. *et al.* Different glacier status with atmospheric circulations in Tibetan Plateau and surroundings. *Nature Climate Change* **2**, 663–667 (2012). <https://doi.org/10.1038/nclimate1580>
- 3 Qu, B., Zhang, Y., Kang, S. & Sillanpää, M. Water quality in the Tibetan Plateau: Major ions and trace elements in rivers of the "Water Tower of Asia". *Sci. Total Environ.* **649**, 571–581 (2019). <https://doi.org/10.1016/j.scitotenv.2018.08.316>
- 4 Liu, Y. *et al.* A genome and gene catalog of glacier microbiomes. *Nat. Biotechnol.* **40**, 1341–1348 (2022). <https://doi.org/10.1038/s41587-022-01367-2>
- 5 Kong, W. *et al.* Autotrophic microbial community succession from glacier terminus to downstream waters on the Tibetan Plateau. *FEMS Microbiol. Ecol.* **95**, fiz190 (2019). <https://doi.org/10.1093/femsec/fiz074>
- 6 Ji, M. *et al.* Salinity reduces bacterial diversity, but increases network complexity in Tibetan Plateau lakes. *FEMS Microbiol. Ecol.* **95**, fiz190 (2019). <https://doi.org/10.1093/femsec/fiz190>
- 7 Zhu, X. *et al.* Vertical variations in microbial diversity, composition, and interactions in freshwater lake sediments on the Tibetan plateau. *Front. Microbiol.* **14**, 1118892 (2023). <https://doi.org/10.3389/fmicb.2023.1118892>
- 8 Guo, L. *et al.* Temperature governs the distribution of hot spring microbial community in three hydrothermal fields, Eastern Tibetan Plateau Geothermal Belt, Western China. *Sci. Total Environ.* **720**, 137574 (2020). <https://doi.org/10.1016/j.scitotenv.2020.137574>
- 9 Upin, H. E., Newell, D. L., Colman, D. R. & Boyd, E. S. Tectonic settings influence the geochemical and microbial diversity of Peru hot springs. *Communications Earth & Environment* **4**, 112 (2023). <https://doi.org/10.1038/s43247-023-00787-5>
- 10 Beghini, F. *et al.* Integrating taxonomic, functional, and strain-level profiling of diverse microbial communities with bioBakery 3. *Elife* **10**, e65088 (2021). <https://doi.org/10.7554/eLife.65088>
- 11 Buchfink, B., Xie, C. & Huson, D. H. Fast and sensitive protein alignment using DIAMOND. *Nat. Methods* **12**, 59–60 (2015). <https://doi.org/10.1038/nmeth.3176>
- 12 Guo, J. *et al.* VirSorter2: a multi-classifier, expert-guided approach to detect diverse DNA and RNA viruses. *Microbiome* **9**, 37 (2021). <https://doi.org/10.1186/s40168-020-00990-y>
- 13 Nayfach, S., Shi, Z. J., Seshadri, R., Pollard, K. S. & Kyrpides, N. C. New insights from uncultivated genomes of the global human gut microbiome. *Nature* **568**, 505–510 (2019). <https://doi.org/10.1038/s41586-019-1058-x>
- 14 Parks, D. H., Imelfort, M., Skennerton, C. T., Hugenholtz, P. & Tyson, G. W. CheckM: assessing the quality of microbial genomes recovered from isolates, single cells, and metagenomes. *Genome Res.* **25**, 1043–1055 (2015). <https://doi.org/10.1101/gr.186072.114>
- 15 Nawrocki, E. P. & Eddy, S. R. Infernal 1.1: 100-fold faster RNA homology searches. *Bioinformatics* **29**, 2933–2935 (2013). <https://doi.org/10.1093/bioinformatics/btt509>
- 16 Bowers, R. M. *et al.* Minimum information about a single amplified genome (MISAG) and a metagenome-assembled genome (MIMAG) of bacteria and archaea. *Nat. Biotechnol.* **35**, 725–731 (2017). <https://doi.org/10.1038/nbt.3893>
- 17 Olm, M. R., Brown, C. T., Brooks, B. & Banfield, J. F. dRep: a tool for fast and accurate

- genomic comparisons that enables improved genome recovery from metagenomes through de-replication. *Isme j* **11**, 2864–2868 (2017). <https://doi.org:10.1038/ismej.2017.126>
- 18 Chaumeil, P. A., Mussig, A. J., Hugenholtz, P. & Parks, D. H. GTDB-Tk v2: memory friendly classification with the genome taxonomy database. *Bioinformatics* **38**, 5315–5316 (2022). <https://doi.org:10.1093/bioinformatics/btac672>
- 19 Kalvari, I. *et al.* Rfam 13.0: shifting to a genome-centric resource for non-coding RNA families. *Nucleic Acids Res.* **46**, D335–d342 (2018). <https://doi.org:10.1093/nar/gkx1038>
- 20 Wei, C. *et al.* Metagenomics revealing molecular profiles of microbial community structure and metabolic capacity in Bamucuo lake, Tibet. *Environ. Res.* **217**, 114847 (2023). <https://doi.org:10.1016/j.envres.2022.114847>
- 21 Yun, J. *et al.* Revealing the community and metabolic potential of active methanotrophs by targeted metagenomics in the Zoige wetland of the Tibetan Plateau. *Environ. Microbiol.* **23**, 6520–6535 (2021). <https://doi.org:10.1111/1462-2920.15697>
- 22 Hu, J. *et al.* Insight into co-hosts of nitrate reduction genes and antibiotic resistance genes in an urban river of the qinghai-tibet plateau. *Water Res.* **225**, 119189 (2022). <https://doi.org:10.1016/j.watres.2022.119189>
- 23 Zhao, Z., Zhang, Y., Liu, L., Liu, F. & Zhang, H. Recent changes in wetlands on the Tibetan Plateau: A review. *Journal of Geographical Sciences* **25**, 879–896 (2015). <https://doi.org:10.1007/s11442-015-1208-5>
- 24 Matsen, F. A., Kodner, R. B. & Armbrust, E. V. pplacer: linear time maximum-likelihood and Bayesian phylogenetic placement of sequences onto a fixed reference tree. *BMC Bioinformatics* **11**, 538 (2010). <https://doi.org:10.1186/1471-2105-11-538>
- 25 Parks, D. H. *et al.* A complete domain-to-species taxonomy for Bacteria and Archaea. *Nat. Biotechnol.* **38**, 1079–1086 (2020). <https://doi.org:10.1038/s41587-020-0501-8>
- 26 Whitfield, J. Biogeography. Is everything everywhere? *Science* **310**, 960–961 (2005). <https://doi.org:10.1126/science.310.5750.960>
- 27 O'Malley, M. A. 'Everything is everywhere: but the environment selects': ubiquitous distribution and ecological determinism in microbial biogeography. *Stud. Hist. Philos. Biol. Biomed. Sci.* **39**, 314–325 (2008). <https://doi.org:10.1016/j.shpsc.2008.06.005>
- 28 Nayfach, S. *et al.* A genomic catalog of Earth's microbiomes. *Nat. Biotechnol.* **39**, 499–509 (2021). <https://doi.org:10.1038/s41587-020-0718-6>

Reviewer #1 (Remarks to the Author):

In this manuscript, Cheng and co-authors make public an exciting dataset from the Tibetan Plateau. This resource is fantastic and it will for sure be highly used and cited. The manuscript is well written and all comments from previous reviews were addressed, and I have no further comments.

Reviewer #2 (Remarks to the Author):

The manuscript has improved substantially after revision. Methodological concerns have been addressed, and efforts have been made to include additional external reference data.

Please correct the typo at Line 101: "Acinomycetota" (to Actinomycetota).

Reviewer #3 (Remarks to the Author):

With some minor changes noted below, I support publication of this article.

1: Should read: "... of the Tibetan Plateau's..."

30: I believe the term is "genomospecies", as in a species identified through genomic sequence only. This needs to be fixed throughout.

206: "the lowest count of MAGs" of the 11 coloured phyla, right?

389: Should read: "... expands the number of species present in the GTDB..."

Figure 1: Change "Weland" in bottom right to "Wetland"

Figure 2: I appreciate that you have added the total unique species count to line 172 (9,698), but I think it would be helpful in Figure 2 as well.

Figure 3: The base of plot a is cutoff.

Figure S6 panel b: Pseudomonadota is misspelled.

Figure S7: I intended for Pielou's evenness to be an additional diversity comparison, not a replacement. Can you report both Shannon's diversity and Pielou's evenness?

Reviewer #1 (Remarks to the Author):

In this manuscript, Cheng and co-authors make public an exciting dataset from the Tibetan Plateau. This resource is fantastic and it will for sure be highly used and cited. The manuscript is well written and all comments from previous reviews were addressed, and I have no further comments.

Answer: Many thanks for your comments. Your previous comments have been instrumental in refining the manuscript.

Reviewer #2 (Remarks to the Author):

The manuscript has improved substantially after revision. Methodological concerns have been addressed, and efforts have been made to include additional external reference data.

Please correct the typo at Line 101: "Acinomycetota" (to Actinomycetota).

Answer: Many thanks for your comments. Your prior comments have been instrumental in refining the manuscript. We have addressed the typo "Acinomycetota" and corrected it to "Actinomycetota." (Line 93)

[Main text]

Line 93: Pseudomonadota, Bacteroidota, and Actinomycetota.

Reviewer #3 (Remarks to the Author):

With some minor changes noted below, I support publication of this article.

1: Should read: "... of the Tibetan Plateau's..."

Answer: Many thanks for your comments. We have improved the title according to your and the editors' suggestions (Line 1)

[Main text]

Line 1: A genome and gene catalog of the aquatic microbiomes of the Tibetan Plateau

30: I believe the term is "genomospecies", as in a species identified through genomic sequence only. This needs to be fixed throughout.

Answer: Many thanks for your comments. We have fixed this typo throughout the manuscript. The editor then suggested that we state that species delineation was genome-based and that the term "species" will be used for conciseness early in the text (Line 152 to 153):

[Main text]

Line 150 to 153: Furthermore, we clustered 32,355 MAGs into 10,723 representative genome-based species, using a whole-genome average nucleotide identity (ANI) threshold of 95% and an aligned fraction (AF) threshold of 30% (Fig. 2c and Supplementary Data 1). Delineation of these species was genome-based and the term “species” will be used for conciseness.

206: “the lowest count of MAGs” of the 11 coloured phyla, right?

Answer: Right! Here we only discussed the top 11 phyla containing more than 1% of the total MAGs.

389: Should read: “... expands the number of species present in the GTDB...”

Answer: Many thanks for your comments. We have improved the sentence (**Lines 383 to 384**).

Lines 383 to 384: The TPMC significantly expands the number of species present in the GTDB by 11.5%

Figure 1: Change “Weland” in bottom right to “Wetland”

Answer: Many thanks for your comments. We have revised the typo (**Figure 1d**).

Figure 2: I appreciate that you have added the total unique species count to line 172 (9,698), but I think it would be helpful in Figure 2 as well.

Answer: Many thanks for your comments. We have added this information to **Figure 2c**.

Figure 3: The base of plot a is cutoff.

Answer: Many thanks for your comments. We have fixed the plot (Figure 3).

Figure S6 panel b: Pseudomonadota is misspelled.

Answer: Many thanks for your comments. We have fixed the typo (Figure S6B).

Figure S7: I intended for Pielou's evenness to be an additional diversity comparison, not a replacement. Can you report both Shannon's diversity and Pielou's evenness?

Answer: Many thanks for your comments. We have reported both Shannon's diversity and Pielou's evenness in the revised manuscript (Figure S7).